# β-1,6-Glucan plays a central role in the structure and remodeling of the bilaminate fungal cell wall

Clara Bekirian[1], Isabel Valsecchi[2], Sophie Bachellier-Bassi[1], Cyril Scandola[3], J Inaki Guijarro[4], Murielle Chauvel[1], Thierry Mourer[5], Neil AR Gow[6], Vishu Kumar Aimanianda[7], Christophe d'Enfert[1], Thierry Fontaine[1]*

[1]Institut Pasteur, Université Paris Cité, INRAE, USC2019, Unité Biologie et Pathogénicité Fongiques, Paris, France; [2]EA DYNAMYC 7380, Faculté de Santé, Université Paris-Est Créteil (UPEC), École Nationale Vétérinaire d'Alfort (EnvA), USC Anses, Créteil, France; [3]Institut Pasteur, Université Paris Cité, Ultrastructural Bioimaging Unit, Paris, France; [4]Institut Pasteur, Université Paris Cité, CNRS UMR3528, Biological NMR and HDX-MS Technological Platform, Paris, France; [5]Institut Pasteur, Advanced Molecular Virology Group, Paris, France; [6]Medical Research Council Centre for Medical Mycology, University of Exeter, Exeter, United Kingdom; [7]Institut Pasteur, Université Paris Cité, Immunobiology of Aspergillus Group, Paris, France

*For correspondence:
thierry.fontaine@pasteur.fr

Competing interest: The authors declare that no competing interests exist.

## eLife Assessment

The article will be of broad interest to fungal biologists and fungal immunologists seeking to understand the biosynthesis of the fungal cell wall, in particular of ß-1,6-glucan synthesis and the importance of this so far understudied constituent of the cell wall for cell wall integrity and immune response. The study is of **fundamental** significance and adds structural clarity to the genetic and biochemical basis of this difficult-to-analyze carbohydrate. It opens the potential for understanding its role in immune recognition and potentially as a drug target. Overall, the data is **compelling**, properly controlled, and analyzed.

## Abstract

The cell wall of human fungal pathogens plays critical roles as an architectural scaffold and as a target and modulator of the host immune response. Although the cell wall of the pathogenic yeast *Candida albicans* is intensively studied, one of the major fibrillar components in its cell wall, β-1,6-glucan, has been largely neglected. Here, we show that β-1,6-glucan is essential for bilayered cell wall organization, cell wall integrity, and filamentous growth. For the first time, we show that β-1,6-glucan production compensates the defect in mannan elongation in the outer layer of the cell wall. In addition, β-1,6-glucan dynamics are also coordinated by host environmental stimuli and stresses with wall remodeling, where the regulation of β-1,6-glucan structure and chain length is a crucial process. As we point out that β-1,6-glucan is exposed at the yeast surface and modulate immune response, β-1,6-glucan must be considered a key factor in host–pathogen interactions.

## Introduction

Pathogenic fungi are a significant burden to public health, recently estimated to cause 2.5 million annually attributable deaths worldwide (*Denning, 2024*). Among these, *Candida* species are responsible

for more than 100 million mucosal infections and 1.5 million systemic infections, representing the fourth most frequent cause of bloodstream infection in hospitalized patients (*Denning, 2024*; *Brown et al., 2012*; *Sharma and Chakrabarti, 2023*). *Candida albicans* is the most common opportunistic species, which in healthy individuals often colonizes mucosal surfaces (oral cavity, gastrointestinal tract, as well as genital tract) (*d'Enfert et al., 2021*), but leads to invasive infection, post trauma or when immune system of the host is compromised. *C. albicans* is classified in the critical priority group of the WHO list of fungal pathogens (*WHO, 2022*).

Fungal cells are enveloped by a cell wall, which is essential for protection, integrity, and to maintain cell morphology. During commensalism with human hosts, the cell wall enables fungi to survive host defenses, whereas during pathogenic interactions the cell wall also plays a central role in the polarized growth of invasive hyphae, thereby facilitating host tissue invasion (*Hall, 2015*; *Mishra et al., 2022*). The cell wall architecture is dynamic and therefore critical in modulating the host immune responses (*Brown, 2011*; *Netea et al., 2015*), which are driven by host immune receptors (pattern recognition receptors [PRRs]) (*Netea et al., 2006*; *Erwig and Gow, 2016*; *Dambuza et al., 2017*). Recognition of fungal pathogen-associated molecular patterns (PAMPs) by host PRRs triggers an array of antifungal responses, including phagocytosis, the production of reactive oxygen/nitrogen species (ROS/RNS), neutrophil extracellular traps, and the release of cytokines and chemokines that recruit immune cells to the site of infection and activate adaptive immunity (*Dambuza et al., 2017*; *Lionakis and Levitz, 2018*; *Urban et al., 2006*; *Gow et al., 2011*). Being extracellular, thereby first to interact with the host immune system, fungal cell wall components function as the major PAMPs.

The cell wall of *C. albicans* is a complex structure with a fibrillar outer layer of mannoproteins and an inner polysaccharide layer composed of chitin, β-1,3-glucan and β-1,6-glucan, which are not present in mammalian cells (*Gow and Hube, 2012*). Dectin-1, a C-type lectin receptor that recognizes fungal cell wall β-1,3-glucan, has been demonstrated to play a dominant pro-inflammatory role in antifungal immunity (*Brown et al., 2002*; *Brown and Gordon, 2003*; *Taylor et al., 2007*; *Ferwerda et al., 2009*; *Marakalala et al., 2013*). Chitin particles derived from the *C. albicans* cell wall lead to the selective secretion of the anti-inflammatory cytokine IL-10 (*Wagener et al., 2014*). Mannans from *C. albicans* are recognized by multiple receptors expressed on different immune cells (Mannose receptor, DC-SIGN, TLR4, Dectin-2, Dectin-3), which are involved in yeast phagocytosis and appear to play a critical role in the induction of Th17 responses (*Hall and Gow, 2013*; *Choudhury et al., 2023*). *C. albicans* has developed strategies to sense environmental signals encountered within the host and to regulate the exposure of its cell wall PAMPs. For example, low levels of β-1,3-glucan exposure by *C. albicans* correlate with enhanced colonization of the gastrointestinal tract (*Sem et al., 2016*; *Hopke et al., 2018*) and β-1,3-glucan is mainly hidden from immune recognition by the outer layer of mannan fibrils (*Erwig and Gow, 2016*; *Lenardon et al., 2020*; *Graus et al., 2018*). Although these *C. albicans* cell wall polysaccharide PAMPs have been investigated in the context of host–fungal interaction, the immunomodulatory potentials of β-1,6-glucan, another major constituent of the *C. albicans* cell wall, remains under-investigated.

The composition and organization of the fungal cell wall result from an equilibrium between syntheses and remodeling of the constituent polysaccharides (*Hall and Gow, 2013*; *Gow et al., 2017*). β-Glucans and chitin, which account for about 60% and 2–10% (depending on the morpho-type), respectively, are cross-linked to form the skeletal armor of the *C. albicans* cell wall (*Gow et al., 2011*; *Klis et al., 2009*). Mannoproteins, which are predominantly glycosylphosphatidylinositol (GPI)-modified and covalently linked to the skeletal polysaccharides via β-1,6-glucan bridges (*Klis et al., 2009*; *Liu et al., 2016*), account for 30–40% of the cell wall. β-1,6-Glucan is composed of a linear chain of β-1,6-linked glucose residues with few short side chains of β-1,3-glucoside or laminaribioside residues (*Iorio et al., 2008*; *Aimanianda et al., 2009*). According to analytical methods, the β-1,6-glucan content is highly variable, from 15 to 50% (*Mio et al., 1997*; *Herrero et al., 2004*; *Free, 2013*; *Schiavone et al., 2014*). Although β-1,6-glucan plays a crucial structural role in the *C. albicans* cell wall, the dynamics of this cell wall polysaccharide as a function of growth or stress conditions remains underexplored.

Genes required for the full β-1,6-glucan production have been identified in model yeast, *Saccharomyces cerevisiae* (*Brown et al., 1993*). Among them, *KRE5* codes a putative α-glucosyltransferase (*CAZy, 2024a*) (GT-24) localized in the endoplasmic reticulum (ER); its deletion results in the absence of β-1,6-glucan in *S. cerevisiae* and about 80% reduction in cell wall β-1,6-glucan content in *C. albicans*

(*Herrero et al., 2004*; *Levinson et al., 2002*). Functions of Kre1, Kre6, Kre9 and the related homologs are unclear. Their localization in the ER, Golgi, plasma membrane, or extracellular culture medium suggests an intracellularly initiating biosynthetic pathway of β-1,6-glucan (*Lesage and Bussey, 2006*; *Montijn et al., 1999*). *KRE6* and its functional homolog *SKN1* share 66% identity and both encode transmembrane proteins containing a glycosyl-hydrolase domain (GH-16). *KRE6* deletion in *S. cerevisiae* results in a 70% decrease in the cell wall β-1,6-glucan content (*Aimanianda et al., 2009*). In *C. albicans,* the double deletion of *KRE6* and *SKN1* leads to a slow growth, cell wall abnormalities, cell separation failure, and reduced virulence (*Han et al., 2019a*; *Han et al., 2019b*). Two more *KRE6* homologs (*KRE62* and *SKN2*) have been identified in *C. albicans* (*CGD, 2024*) but their function has not yet been investigated, and the *C. albicans* cell wall β-1,6-glucan biosynthetic pathway, namely the β-1,6-glucosyltransferase involved in β-1,6-glucan polymerization and other enzymes involved in β-1,6-glucan remodeling (elongation, branching, or cross-linking with β-1,3-glucan), remains to be identified.

We standardized biochemical approaches to quantify as well as to characterize β-1,6-glucan of *C. albicans*, which allowed us to perform comparative analysis and to study the dynamic of this cell wall polysaccharide in response to stress conditions and to antifungal treatments. β-1,6-Glucan biosynthesis appears to be a compensatory reaction when mannan elongation is defective, whereas the absence of β-1,6-glucan results in a significantly sick growth phenotype and complete cell wall reorganization. Moreover, β-1,6-glucan showed a distinct immunomodulatory response compared to other cell wall polysaccharides. Together, our study establishes overarching and critical roles for β-1,6-glucans in the organization, structure, biosynthesis, and immunomodulatory properties of the cell wall.

## Results

### β-1,6-Glucan: A critical polysaccharide of *C. albicans* cell wall

Despite numerous studies (*Mio et al., 1997*; *Herrero et al., 2004*; *Free, 2013*; *Schiavone et al., 2014*), the composition and organization of *C. albicans* cell wall β-1,6-glucan remain unclear. To better understand the composition and organization of the cell wall β-1,6-glucan, we have fractionated cell wall and applied new biochemical strategies to quantify and characterize them. The *C. albicans* cell wall was separated into three fractions according to polymer properties and reticulations, namely sodium-dodecyl-sulfate-β-mercaptoethanol (SDS-β-ME) extract, alkali-insoluble (AI), and alkali-soluble (AS) fractions (see 'Materials and methods'). In synthetic dextrose (SD) medium at 37°C, the SDS-β-ME fraction was the smallest (18% of total cell wall) and mainly composed of mannoproteins (*Figure 1a*). The AI fraction was the major one (46%) and was exclusively composed of glucose and glucosamine, the monosaccharide building blocks of β-glucans and chitin, respectively. The AS fraction (36% of total cell wall) was characterized by a high amount of mannose residues (54% of the fraction) resulting from the release of covalently cell wall-anchored mannoproteins (*Figure 1a*), and this fraction also contained glucose residues originating from β-glucans. Overall, we found that the *C. albicans* cell wall is composed of 3% of chitin, 43% of proteins/mannoproteins (27% of mannans), and 54% of β-glucans, which was in agreement with earlier studies and validated our analytic approach (*Figure 1a and b*; *Chaffin et al., 1998*; *Garcia-Rubio et al., 2020*; *Ruiz-Herrera et al., 2006*).

To differentiate β-1,3-glucans and β-1,6-glucans, we developed two methods, one for the AI fraction and other for the AS fraction (see 'Materials and methods' and *Figure 1—figure supplement 3*) as β-glucan was found in both fractions. β-1,6-Glucan corresponded to 23% of the total cell wall and was mainly found in the AI fraction (*Figure 1a and b*). β-1,6-Glucan in the AI fraction could be solubilized by digesting β-1,3-glucan using an endo-β-1,3-glucanase, confirming that it is covalently linked to the cell wall β-1,3-glucan as previously shown in *S. cerevisiae* (*Aimanianda et al., 2009*; *Kollár et al., 1997*). Further, gel filtration chromatography showed that the solubilized β-1,6-glucan was eluted in a single symmetric peak with Gaussian shape showing a homogeneous polymer with an apparent molecular weight range of 18–112 kDa (*Figure 1d*), and the average molecular weight was estimated to be 58 kDa. The structure of purified β-1,6-glucan was analyzed by $^1$H and $^{13}$C-NMR (*Figure 1—figure supplement 1*) and by high-performance anion exchange chromatography (HPAEC) after endo-β-1,6-glucanase digestion (*Figure 1e*). The nuclear magnetic resonance (NMR) chemical shifts and coupling constants as well as the HPAEC profile of the digested products were highly comparable to those we described for *S. cerevisiae* (*Aimanianda et al., 2009*), showing a

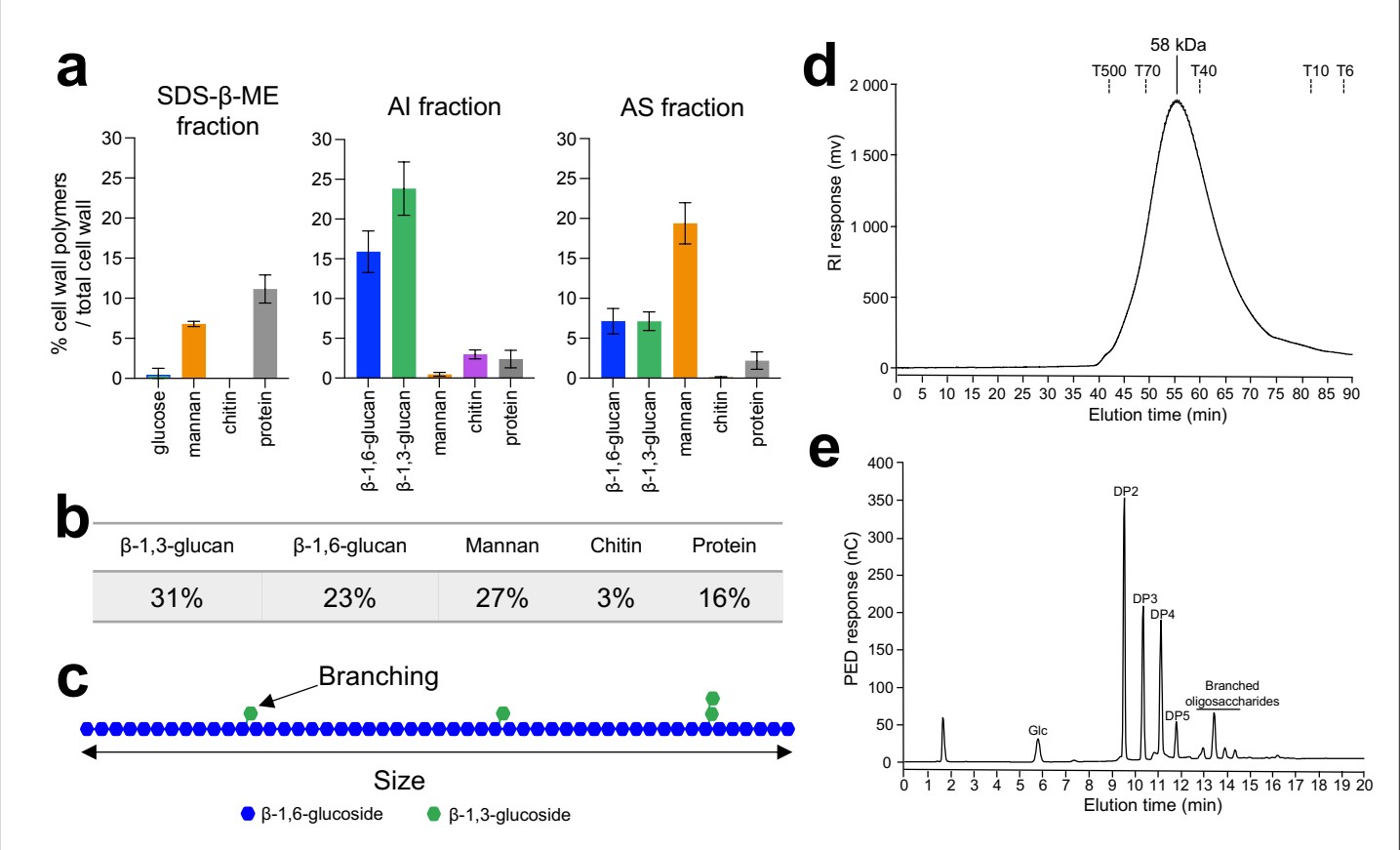

**Figure 1.** Analysis of *C. albicans* cell wall β-1,6-glucans. (**a**) Percentages of cell wall polymers on total cell wall, distributed by fractions: sodium-dodecyl-sulfate-β-mercaptoethanol (SDS-β-ME), alkali-insoluble (AI), and alkali-soluble (AS). Cells were grown in synthetic dextrose (SD) medium at 37°C. Means and standard deviations were calculated from three independent experiments. (**b**) Table of the mean percentages of each polymer in the cell wall from three independent experiments. (**c**) Diagram of β-1,6-glucan structure. In blue are represented glucose residues linked in β-1,6 and in green glucose residues linked in β-1,3. According to nuclear magnetic resonance (NMR) analysis and high-performance anion exchange chromatography (HPAEC) after endo-β-1,6-glucanase digestion (*Figure 1—figure supplement 1*), based on three independent experiments, an average of 6.4% (± 0.5%) of glucose units of the main chain are substituted by a single glucose residue (88–90%) or a laminaribiose (10–12%). (**d**) Gel filtration analysis on a Superdex 200 column of β-1,6-glucan released by endo-β-1,3-glucanase digestion. The column was calibrated with dextrans (Tx: × kDa). (**e**) HPAEC analysis of the digestion products of the AI fraction treated with an endo-β-1,6-glucanase. Chromatographs in (**d**) and (**e**) are representative of three independent experiments. PED, pulsed electrochemical detector; nC, nanocoulombs; RI, refractive index; mV, millivolt; DP, degree of polymerization; Glc, glucose.

The online version of this article includes the following source data and figure supplement(s) for figure 1:

**Source data 1.** Raw data for panels a and b.

**Figure supplement 1.** $^{1}$H and $^{13}$C NMR resonance assignments, $^{3}J_{H1/H2}$ and $^{1}J_{H1/C1}$ coupling constants of the monosaccharide residues of cell wall β-1,6-glucan purified from the alkali-insoluble (AI) fraction.

**Figure supplement 2.** Cell disruption is essential to eliminate glycogen in alkali-insoluble (AI) and alkali-soluble (AS) fractions.

**Figure supplement 3.** Quantification methods of β-1,6-glucans in alkali-insoluble (AI) fractions.

similar structure: a main chain of β-1,6-glucosyl residues with side chains of β-1,3-glucosyl or β-1,3-laminaribiosyl residues. The relative intensity of the anomeric proton in NMR and HPAEC analysis after endo-β-1,6-glucanase digestion indicated that an average of 6.4% of glucose units of the main chain is substituted and that 88–90% of the side chains is composed of a single glucose residue (*Figure 1c and e*, *Figure 1—figure supplement 1*). Altogether, our data showed that β-1,6-glucan is a major cell wall polymer in *C. albicans*, representing 40% of the cell wall β-glucans.

## Environmental regulation of β-1,6-glucan synthesis

The composition and surface exposure of fungal cell wall polymers are dependent on environmental growth conditions (*Hopke et al., 2018*; *Gow et al., 2017*). We applied chemical assays and our

biochemical approaches to enable comparative analyses of cell wall robustness in relation to β-1,6-glucan dynamics in *C. albicans*.

First, we analyzed the cell wall composition in relation to cell morphotype and growth phase (exponential vs. stationary phase, yeast vs. hyphae, planktonic vs. biofilm). No significant difference was observed between cell wall β-1,6-glucan and other polymers produced during the exponential (control) and stationary phases in yeast cells (*Figure 2*). In biofilms, only a slight increase in mannan content (26% vs. 20%), but no change in relative β-glucan content, was observed. In contrast, the cell wall isolated from *C. albicans* hyphae was characterized by an increase in chitin content (10% vs. 3%) as previously shown (*Gow et al., 2017*; *Ruiz-Herrera et al., 2006*; *Braun and Calderone, 1978*), as well as a reduction in β-1,6-glucan (18% vs. 23%) and mannan contents (13% vs. 20%) compared to the yeast form. The decrease in β-1,6-glucan content was reflected in a reduction in the β-1,6-glucan size (43 vs. 58 kDa), demonstrating that both content and structure of β-1,6-glucan are regulated during growth and cellular morphogenesis. In addition, a higher branching rate of β-1,3-glucan of the hyphal cell wall was also observed in the AI fraction (7.3% vs. 5.3%) (*Figure 2—figure supplement 2*).

We further investigated the effect on cell wall composition of environmental growth conditions, such as temperature, pH, carbon source, and hypoxia, on cell wall composition (*Figure 2*). Hypoxic condition (1% $O_2$, 5% $CO_2$) resulted in a slow growth but did not lead to a significant alteration of the cell wall composition. In contrast, all other tested conditions led to a modification of cell wall composition including the proportion of β-1,6-glucan. For example, growth at 30°C or in 1 M NaCl led to a decrease in the cell wall β-1,6-glucan content and a decrease in β-1,6-glucan chain length. However, these decreases at low temperature were compensated by a higher content of less-branched β-1,3-glucan, whereas high salt led to an increased branching rate of both β-1,6-glucan and β-1,3-glucan (*Figure 2*, *Figure 2—figure supplement 2*). Buffered culture medium at pH 4 or 7.5 led to a slight decrease in the β-1,6-glucan molecular weight and an increase in its branching frequency in comparison to buffered medium at pH 5.4. However, the most pronounced alteration of the cell wall was observed in lactate containing medium, with a steep reduction in polysaccharides from the structural core, chitin (−43%), β-1,3-glucan (−48%), and mainly β-1,6-glucan (−72%), in agreement with the decrease in thickness of the inner cell wall layer previously described (*Ene et al., 2012*). Lactate medium also led to a strong reduction in the average β-1,6-glucan molecular weight (19 vs. 58 kDa), a decrease in β-1,6-glucan branching (4.9 vs. 6.4%), and an increase in β-1,3-glucan branching (7.4 vs. 5.3%, *Figure 2—figure supplement 2*).

The presence in the medium of short-chain fatty acids (acetate, propionate, or butyrate) at physiological gut concentrations resulted in a growth defect but did not alter the global cell wall composition. The main difference was an increase in β-1,6-glucan molecular weight (64–68 kDa vs. 58 kDa) (*Figure 2b*). Oxidative stress and drugs targeting the cell wall did not induce any major alterations (*Figure 2*). Cell wall chitin content was reduced in the presence of nikkomycin, a chitin synthesis inhibitor, and increased in the presence of the cell wall-damaging agents Congo Red or caspofungin, as previously observed (*Chapman et al., 1992*; *Roncero and Durán, 1985*; *Walker et al., 2013*), but with no impact on β-1,3-glucan and β-1,6-glucan contents. The major stress effect we observed was in the presence of caspofungin, which caused an increase in the average molecular weight of β-1,6-glucan (70 vs. 58 kDa).

These data demonstrate that the content, size, and branching of β-1,6-glucan in the *C. albicans* cell wall are regulated by numerous environmental growth and stress conditions.

## β-1,6-Glucan dynamics responds to alterations in inner cell wall synthesis

We investigated the dynamics of β-1,6-glucan in *C. albicans* mutants in which the genes involved in the biosynthesis or remodeling of chitin and β-1,3-glucan were deleted. Four chitin synthases, located at the plasma membrane, have been described in *C. albicans*, and we have analyzed three deletion mutants. Chs1 is involved in primary septum formation and is essential for vegetative growth. We therefore analyzed the conditional $P_{MRP1}$-*CHS1*/*chs1*Δ mutant under repressive condition. Chs2 is a nonessential class II enzyme and represents the major chitin synthase activity measured in cell membrane preparations (*Gow et al., 1994*). Chitin synthase 3 (Chs3) is responsible for most of the cell wall chitin production and chitin ring formation at sites of cell division (*Preechasuth et al., 2015*) but it is not essential for growth and viability. Our comparative analyses did not show any alteration in

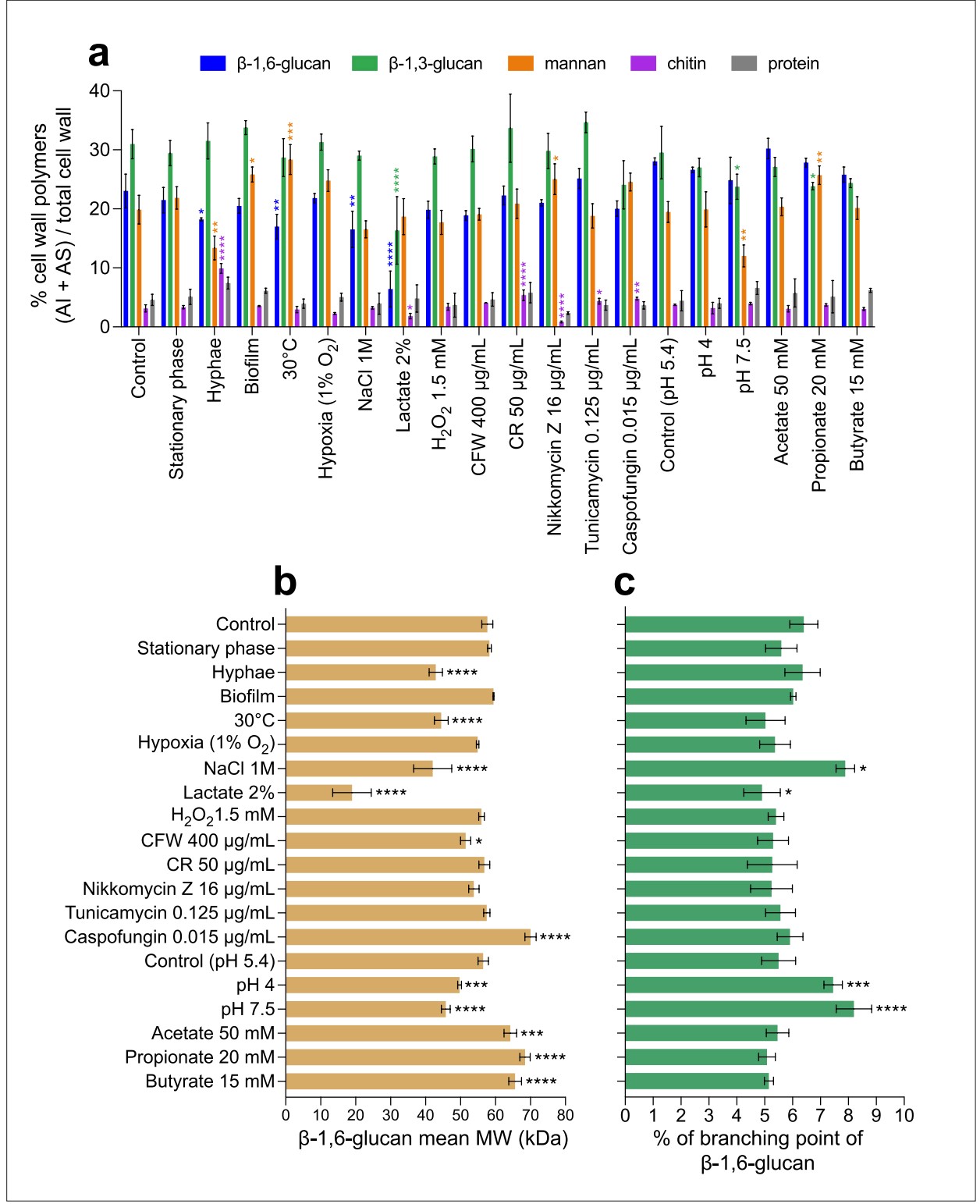

**Figure 2.** Comparative analyses of cell wall β-1,6-glucans produced in various environmental conditions. (**a**) Percentages of cell wall polymers (alkali-insoluble [AI] + alkali-soluble [AS] fractions) on total cell wall. Cells were grown in liquid synthetic medium at 37°C under different conditions, as specified in 'Materials and methods'. (**b**) β-1,6-Glucan mean molecular weight (MW). Average molecular weight was estimated by gel filtration chromatography on a Superdex 200 column. (**c**) Branching rate of β-1,6-glucans. Branching rate was determined by high-performance anion exchange chromatography (HPAEC) after digestion of the AI fraction by an endo-β-1,6-glucanase (% expressed as number of glucose units of the main chain that are substituted by a side chain). For (**a–c**), means and standard deviations from three independent replicate experiments are shown. All data were

*Figure 2 continued on next page*

*Figure 2 continued*

compared to the control conditions and were analyzed by one-way ANOVA with Dunnett's multiple comparisons test: *p<0.05; **p<0.01; ***p<0.001; ****p<0.0001.

The online version of this article includes the following source data and figure supplement(s) for figure 2:

**Source data 1.** Raw data for panels a, b and c.

**Figure supplement 1.** Global cell wall composition produced by *C. albicans* in different environmental conditions.

**Figure supplement 1—source data 1.** Raw data for panels a, b and c.

**Figure supplement 2.** Branching rates of β-1,6-glucans and β-1,3-glucans produced by *C. albicans* under various environmental conditions.

**Figure supplement 2—source data 1.** Raw data for panels a and b.

the chitin content in the P$_{MRP1}$-CHS1/chs1Δ and chs2Δ/Δ mutants, but a strong reduction was observed in the *chs3Δ/Δ* mutant (–92.3%) (***Figure 3—figure supplement 1***) in accordance with previous data (***Munro et al., 1998***; ***Mio et al., 1996***). In the P$_{MRP1}$-CHS1/chs1Δ mutant, the cell wall had a 45.7% decrease in mannan content in the AS fraction and a 50% increase in β-1,6-glucan in the AI fraction (***Figure 3—figure supplement 1***, ***Figure 3a***). In the *chs3Δ/Δ* mutant, the low proportion of chitin led to an increase in both β-1,3-glucan (×1.6) and β-1,6-glucan (×1.6) in the AS fraction (***Figure 3a***, ***Figure 3—figure supplement 1***), suggesting a defect in β-glucans cross-linking in the cell wall polysaccharide core. No significant change in β-1,6-glucan structure was noticed for these two mutants (***Figure 3b and c***). Strikingly, no significant effect of the *CHS2* deletion was observed on cell wall chitin content or overall wall composition.

*FKS1*, *FKS2,* and *FKS3* encode β-1,3-glucan synthases in *C. albicans* (***Ruiz-Herrera et al., 2006***; ***Suwunnakorn et al., 2018***). As *FKS1* is essential for vegetative growth (***Suwunnakorn et al., 2018***), we used the heterozygous *fks1Δ* mutant, which displays a haplo-insufficiency phenotype (***Lee et al., 2018***). Only a decrease (–58.7%) in β-1,3-glucan content in the AS fraction (***Figure 3—figure supplement 1***) was observed, without any other significant changes. Moreover, essentiality of *FKS1* prevented analysis of β-1,6-glucan dynamics in a β-1,3-glucan-deficient mutant. *PHR1* and *PHR2* encode β-1,3-glucanosyltransferases of the GH-72 family involved in β-1,3-glucan elongation and branching at the extracellular cell wall level (***Fonzi, 1999***; ***Aimanianda et al., 2017***). In *C. albicans*, their expression is strongly pH dependent: in vitro, *PHR1* is expressed above pH 5.5, while *PHR2 is* expressed below pH 5.5 (***De Bernardis et al., 1998***). The *phr1Δ/Δ* mutant was thus grown at pH 7.5 and *phr2Δ/Δ* at pH 5.5. As expected for both mutants, defects in GH-72 β-1,3-glucan remodelase led to a decrease in β-1,3-glucan branching (–67.3% for *phr1Δ/Δ*, –31.5% for *phr2Δ/Δ*) (***Figure 3—figure supplement 2***) and a compensatory increase in the cell wall chitin content (1.5× for *phr1Δ/Δ* and 3.7× for *phr2Δ/Δ*) (***Figure 3—figure supplement 1***). No change in the cell wall β-1,3-glucan content was observed in the *phr1Δ/Δ* mutant (***Figure 3—figure supplement 1***). In contrast, deletion of *PHR2* led to an increase in β-1,3-glucan content and a cross-linking defect (–31.5% estimated on the ratio β-1,3-glucan content in AS and AI fractions). Concomitantly, deletion of *PHR2* led to a reduction in cell wall β-1,6-glucan content in both AI (–58.7%) and AS (–51.3%) fractions and to a reduction in molecular weight and branching of the β-1,6-glucan chains (***Figure 3a***). In the *phr1Δ/Δ* mutant, only a significant increase in the molecular weight of β-1,6-glucan chain was observed (60 vs. 50 kDa) (***Figure 3***).

Taken together, these data showed that mutational disruption of chitin-synthase or β-1,3-glucan-synthase genes did not significantly alter the cell wall β-1,6-glucan structure. However, defects in chitin content led to a decrease in the cross-linking of β-glucans in the inner wall (i.e., AI fraction) that corresponds to the effect of nikkomycin-treated *C. albicans* phenotype (***Figure 2—figure supplement 1***); conversely, an increase in chitin content led to more cross-linking of β-glucans as observed in the *FKS1* mutant or in the presence of caspofungin (***Figure 2—figure supplement 1***, ***Figure 3—figure supplement 1***). In contrast, disruption of cell wall β-1,3-glucan remodeling had a pH-dependent effect on β-1,6-glucan structure (***Figures 2 and 3***).

## A β-1,6-glucan mannan compensatory pathway links the regulation of the inner and outer cell wall layers

In *C. albicans*, mannosylation of cell wall proteins is required for the organization of the fibrillar outer layer of the cell wall that is composed of *N*-linked mannans (***Lenardon et al., 2020***; ***Gow and Lenardon, 2023***). This fibrillar layer is anchored to the polysaccharide core composed of β-1,3-glucan

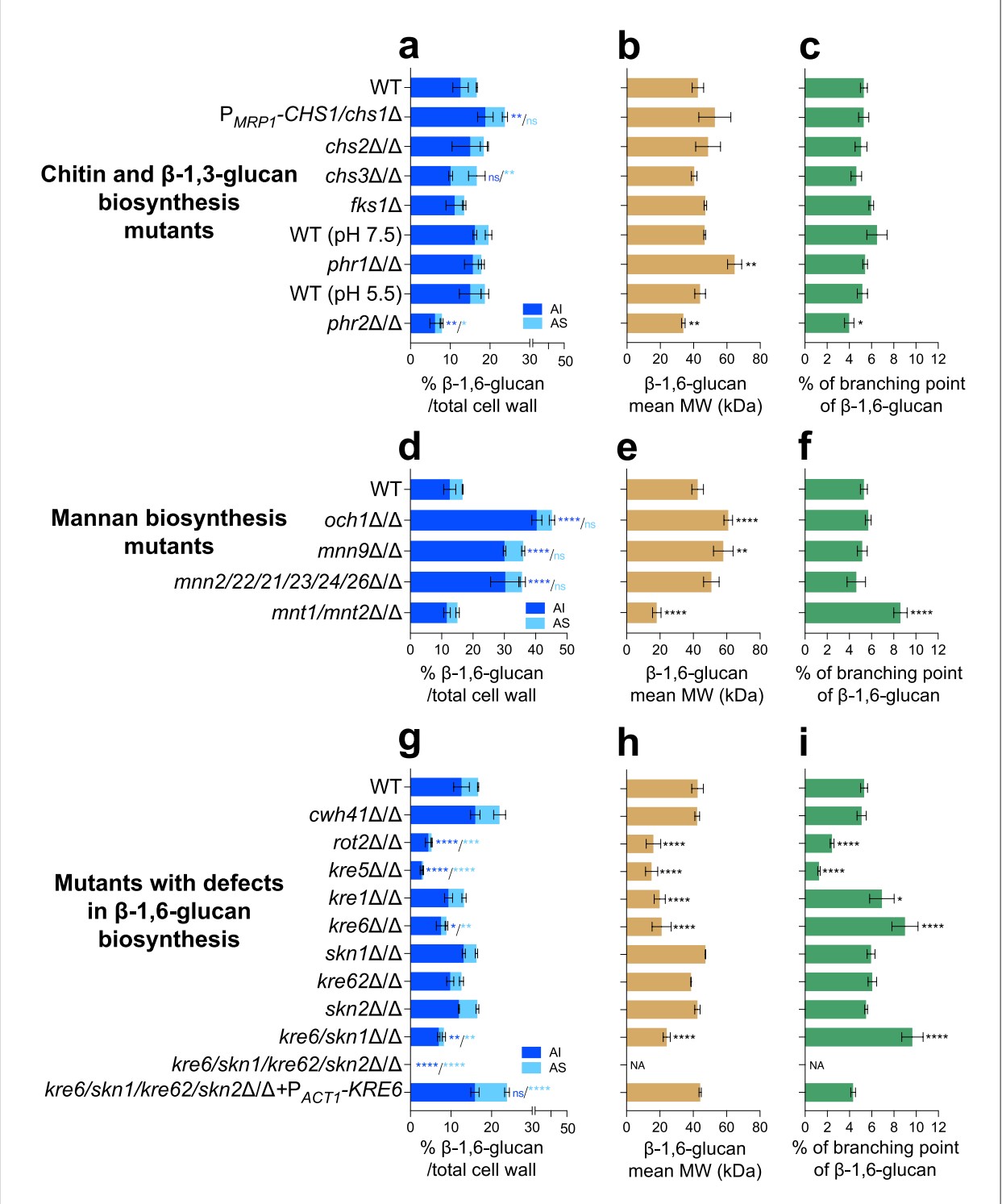

**Figure 3.** Comparative analysis of β-1,6-glucan content and structure produced by cell wall mutants. (**a, d, g**) Percentages of cell wall β-1,6-glucans (alkali-insoluble [AI] and alkali-soluble [AS] fractions) on total cell wall. (**b, e, h**) β-1,6-Glucans mean molecular weight (MW). (**c, f, i**) Branching rate of β-1,6-glucans. Cells were grown in liquid synthetic dextrose (SD) medium at 30°C. Means and standard deviations from three independent replicate experiments are shown. All data were compared to the control conditions and were analyzed by one-way ANOVA with Dunnett's multiple comparisons test: *p<0.05; **p<0.01; ***p<0.001; ****p<0.0001; ns, nonsignificant; NA, nonapplicable.

The online version of this article includes the following source data and figure supplement(s) for figure 3:

**Source data 1.** Raw data for panels a to i.

*Figure 3 continued on next page*

*Figure 3 continued*

**Figure supplement 1.** Cell wall composition of *C. albicans* mutants.

**Figure supplement 1—source data 1.** Raw data for panels a, b and c.

**Figure supplement 2.** Branching rates of β-1,6-glucans and β-1,3-glucans produced by different cell wall mutants of *C. albicans*.

**Figure supplement 2—source data 1.** Raw data for panels a and b.

**Figure supplement 3.** Absence of β-1,6-glucans in the cell wall of the quadruple *kre6/kre62/skn2/skn1Δ/Δ* mutant.

---

and chitin, via β-1,6-glucan bridges (*Klis et al., 2009*). We analyzed mutants deficient in Golgi α-mannosyltransferases involved in *O*- and *N*-mannosylation. Mnt1 and Mnt2 are α-1,2-mannosyltransferases involved in catalyzing consecutive reactions in *O*-mannan elongation (*Hall and Gow, 2013*; *Munro et al., 2005*). As predicted, the cell wall of the *mnt1/mnt2Δ/Δ* mutant showed minor changes in overall mannan content (*Figure 3d*, *Figure 3—figure supplement 1*; *Munro et al., 2005*). However, deletion of *MNT1* and *MNT2* genes led to a strong reduction in β-1,6-glucan molecular weight (18 vs. 43 kDa) and an increased branching of β-1,6-glucan chains (8.6 vs. 5.3%) (*Figure 3e and f*). A slight increase in the β-1,3-glucan branching was also observed in the *mnt1/mnt2Δ/Δ* double mutant (*Figure 3—figure supplement 2*). Overall, these data suggested that β-1,6-glucan and mannan syntheses may be coupled.

*N*-mannan elongation requires the activity of a set α-1,6-mannosyltransferases in the Mannan Polymerase I and II complexes for the formation of α-1,6-mannan backbone that is subsequently modified with α-1,2- and α-1,3-mannosyltransferases that assemble the *N*-mannan side chains (*Hall and Gow, 2013*). Och1 activity is responsible for the initial addition of α-1,6-mannose onto the relatively well conserved *N*-glycan triantennary complex of glycoproteins, and Mnn9 activity is essential for the polymerization of the α-1,6-mannose backbone. Mnn2, Mnn21, Mnn22, Mnn23, Mnn24, and Mnn26 activities are essential to the synthesis of side chains (*Hall and Gow, 2013*). Single *och1Δ/Δ* and *mnn9Δ/Δ* mutants and the sextuple *mnn2/21/22/23/24/26Δ/Δ* mutant resulted in a major decrease in mannan content in the AS fraction (–90.4% for *och1Δ/Δ*, –88.9% for *mnn9Δ/Δ*, and –87.1% for *mnn2/21/22/23/24/26Δ/Δ*) and a compensatory increase in chitin content (×3.1 for *och1Δ/Δ*, ×2.5 for *mnn9Δ/Δ*, and ×2.5 for *mnn2/21/22/23/24/26Δ/Δ*) in the AI fraction (*Figure 3—figure supplement 1*), as previously described (*Hall and Gow, 2013*). A significant increase in cell wall β-1,3-glucan content was observed in the *mnn9Δ/Δ* and *mnn2/21/22/23/24/26Δ/Δ* mutants. There was also a significant increase in the β-1,6-glucan content (×3.2 for *och1Δ/Δ*, ×2.4 for *mnn9Δ/Δ*, and ×2.4 for *mnn2/21/22/23/24/26Δ/Δ*, respectively) concomitant with the elongation of β-1,6-glucan chains, and consequential increase in its molecular weight (*Figure 3d and e*).

Overall, these data suggest that the β-1,6-glucan biosynthesis is stimulated via a compensatory pathway when there is a defect in *O*- and *N*-linked cell wall mannan biosynthesis. This novel compensatory response occurs by increasing the β-1,6-glucan content in *N*-mannan elongation-deficient mutants and increasing branching of shorter β-1,6-glucan chains in *O*-mannan-deficient mutants.

### *KRE6*-family-members-dependent β-1,6-glucan biosynthesis

Genes encoding proteins essential for β-1,6-glucan biosynthesis were first identified in *S. cerevisiae* as mutants resistant to Killer Toxin K1 (*KRE*) (*Brown et al., 1993*; *Shahinian and Bussey, 2000*). In addition, a range of proteins in the endoplasmic reticulum and Golgi (Cwh41, Rot2, Kre5, Kre6, Skn1) are critical for the assembly of this polymer (*Shahinian and Bussey, 2000*). We analyzed homologs of *CWH41*, *ROT2*, *KRE5* in *C. albicans*. These three calnexin cycle proteins involved in glucosylation/deglucosylation of *N*-glycan in the ER (*Aebi et al., 2010*) had a reduced amount of cell wall mannan (*Figure 3—figure supplement 1*). Strong cell wall phenotypes were observed for *kre5Δ/Δ* and *rot2Δ/Δ* mutants, which exhibited significant increases in the highly branched β-1,3-glucan (×1.8 for *rot2Δ/Δ*, ×2.2 for *kre5Δ/Δ*) (*Figure 3—figure supplement 2*), chitin (×4.2 for *rot2Δ/Δ*, ×5.2 for *kre5Δ/Δ*) and a strong decrease in β-1,6-glucan in the *rot2Δ/Δ* (–68.7%) and *kre5Δ/Δ* mutants (–81.5%) (*Figure 3g*, *Figure 3—figure supplement 1*). In addition, the β-1,6-glucan structure was also altered in both mutants, with a decrease in the average molecular weight (15–16 kDa instead of 43 kDa for the parental strain) and a reduction in the side chain branching (–54.2% for *rot2Δ/Δ* and –76.5% for *kre5Δ/Δ*) (*Figure 3h and i*). *KRE1* encodes a plasma membrane protein whose function is not yet clear (*Breinig et al., 2002*; *Breinig et al., 2004*). The *kre1Δ/Δ C. albicans* mutant exhibited a reduced

molecular weight of β-1,6-glucan chains (20 kDa instead of 43 kDa in the parental strain) (**Figure 3h**) but was not affected in overall cell wall composition (**Figure 3—figure supplement 1**).

In *C. albicans,* the *KRE6* family is composed of four homologs: *KRE6, SKN1, KRE62,* and *SKN2.* These genes encode putative trans-β-glycosylases of the GH-16 family for which no biochemical activity has been described to date (**CAZy, 2024b**). To analyze the function of these four proteins in *C. albicans*, we used a transient CRISPR/Cas9 protocol to build the four single-knockout mutants, the double *kre6/skn1Δ/Δ,* and the quadruple *kre6Δ/Δ/kre62/skn2/skn1Δ/Δ* mutants. Among the single mutants, only *kre6Δ/Δ* exhibited a marked alteration in the cell wall composition (**Figure 3—figure supplement 1**), with significantly lower β-1,6-glucan (–46.8%) (**Figure 3g**) and mannan content (–42.8%) (**Figure 3—figure supplement 1**), and higher β-1,3-glucan (×1.9) and chitin (×3.9) content. In addition, the β-1,6-glucan chain length was shorter (–50.5%) and more branched (×1.7) than in the parental strain (**Figure 3h and i**) and β-1,3-glucan was also hyperbranched (×1.8) (**Figure 3—figure supplement 2**). The *kre6/skn1Δ/Δ* mutant phenotype, β-1,6-glucan content, and structure were similar to the single *kre6Δ/Δ* mutant (**Figure 3g–i**). The quadruple *kre6/kre62/skn2/skn1Δ/Δ* mutant did not produce any measurable β-1,6-glucan (**Figure 3g**, **Figure 3—figure supplement 1**), which was confirmed by endo-β-1,6-glucanase digestion of the AI and AS fractions (**Figure 3—figure supplement 3**). The absence of β-1,6-glucan was associated with an increased chitin content (9.5% vs. 2.5%), and highly branched β-1,3-glucan (48% vs. 20% in the parental strain) (**Figure 3—figure supplements 1 and 2**). Reintegration of *KRE6* under the control of the *ACT1* promoter restored the production of β-1,6-glucan with a structure similar to that of the parental strain (**Figure 3g–i**).

Therefore, in *C. albicans,* deletion of all *KRE6* homologous genes led to the absence of β-1,6-glucan production and structural alterations of the cell wall, showing that the β-1,6-glucan biosynthetic pathway is critical to the regulation of cell wall homeostasis.

## *KRE6* regulation of cell wall architecture

The single deletions of *KRE62, SKN1,* and *SKN2* did not affect the growth of *C. albicans* in SD medium nor drug sensitivity (**Figure 4**), whereas the deletion of *KRE6* led to a significant increase in doubling time (×2.4, **Figure 4a and b**). The double *kre6/skn1Δ/Δ* mutant had the same growth rate as *kre6Δ/Δ* (**Figure 4a and b**). However, deletion of all genes of the *KRE6* homologs family (quadruple *kre6/kre62/skn2/skn1Δ/Δ* mutant) resulted in a stronger growth phenotype with a longer lag phase and a higher doubling time (×3.5 vs. parental strain). Cells of the quadruple *kre6/kre62/skn2/skn1Δ/Δ* mutant were bigger than the parental strain (**Figure 4d**). In addition, the single *kre6Δ/Δ* and all multiple mutants aggregated in SD liquid medium and had increased sensitivity to the cell wall perturbing agents Congo Red, Calcofluor White, as well as to the chitin synthase inhibitor nikkomycin Z and the *N*-glycosylation inhibitor tunicamycin. No growth of the quadruple mutant was observed in the presence of either of these drugs. Surprisingly, caspofungin enhanced the growth of mutants bearing *KRE6* gene deletion (**Figure 4c**). Among the single mutants, only *skn1Δ/Δ* was defective for filamentous growth; all multiple mutants also displayed defects in filamentous growth, and the severity of this phenotype was proportional to the number of *KRE6* homologous genes deleted (**Figure 4d**). The reintegration of *KRE6* under P$_{ACT1}$ restored parental strain-like growth kinetics, filamentation, and drug resistance (**Figure 4**) and confirmed the above differences noted in the mutants of *KRE6* and *SKN1* (**Han et al., 2019b**).

Cell wall architecture, observed by transmission electron microscopy, showed that single *skn1Δ/Δ,* *kre62Δ/Δ,* and *skn2Δ/Δ* mutants presented a bilayered structure with a classical fibrillar outer layer similar to the parental strain (**Figure 5**). However, the outer layer of the *skn1Δ/Δ* and *skn2Δ/Δ* mutants was thicker than the WT (*skn1Δ/Δ*, 70 ± 12 nm; *skn2Δ/Δ*, 66 ± 12 nm vs. WT, 53 ± 10 nm). The cell wall of the *kre6Δ/Δ* and *kre6/skn1Δ/Δ* mutants had a heterogeneous and thicker inner layer (*kre6Δ/Δ*, 210 ± 89 nm; *kre6/skn1Δ/Δ*, 216 ± 68 nm vs. WT, 108 ± 23 nm) (**Figure 5**). The external mannoprotein-rich fibrillar layer was absent in the quadruple *kre6/kre62/skn2/skn1Δ/Δ* mutant and the inner layer was thicker and of variable thickness (440 ± 158 nm vs. 108 ± 23 nm). A normal wall architecture was fully restored in the complemented strain.

These results demonstrate that β-1,6-glucan has a pivotal role in determining the architectural arrangement of polysaccharides in the cell wall, which in turn influences growth, drug sensitivity, cell morphology, and filamentation.

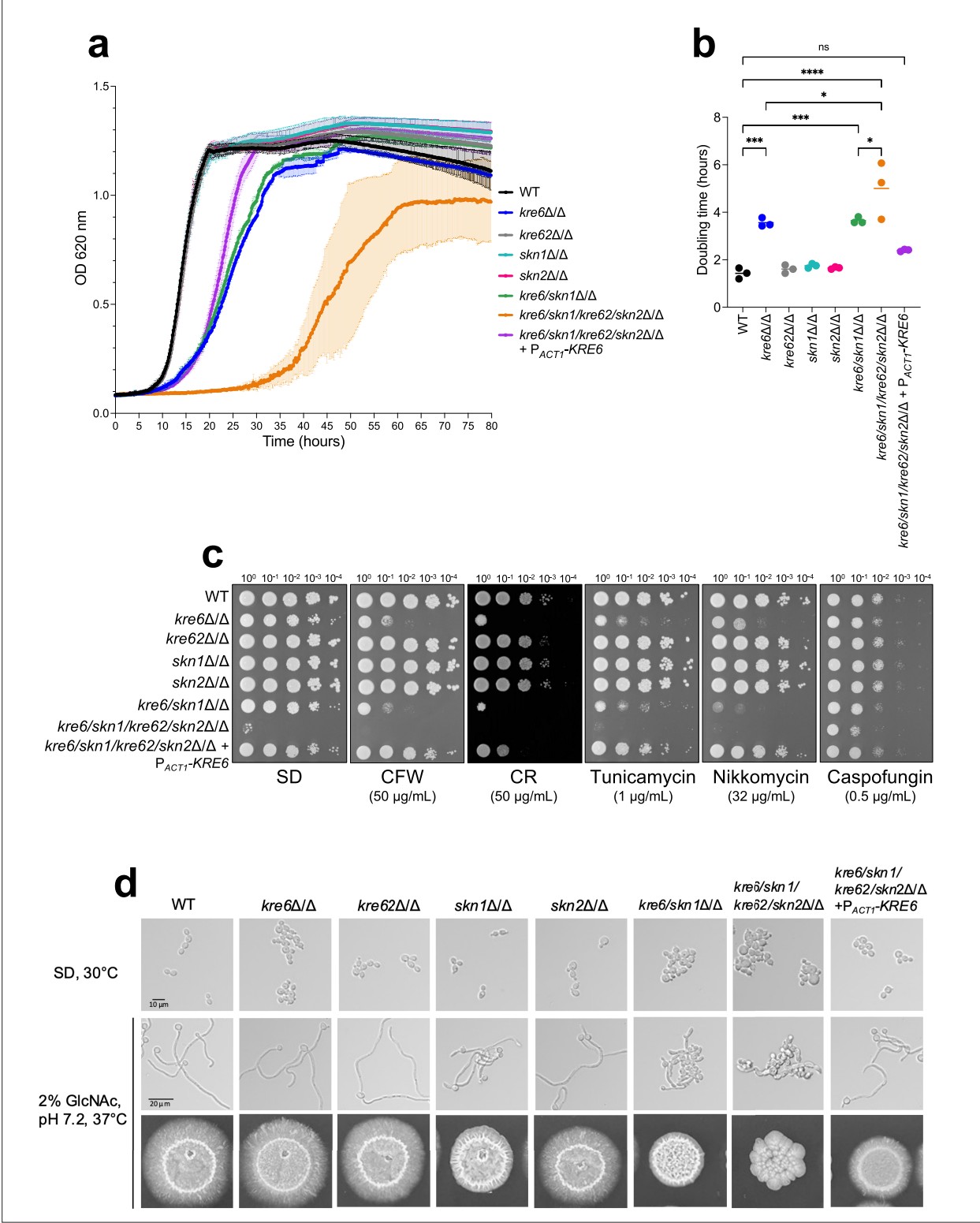

**Figure 4.** Phenotypic characterization of *KRE6* family mutants: growth kinetics, drug susceptibility, and filamentation. (**a**) Kinetic curve of all strains grown in liquid synthetic dextrose (SD) medium, 30°C. Optical density at 620 nm was measured every 10 min during 80 hr by TECAN SUNRISE. Means and standard deviations were calculated from three independent experiments. (**b**) Doubling time of each strain was determined from three independent replicates. Statistical analyses were performed with one-way ANOVA with Tukey's multiple comparisons test: *p<0.05; **p<0.01; ***p<0.001; ****p<0.0001; ns, nonsignificant. (**c**) Spotting test of tenfold serial dilution of yeast cells of all strains on SD medium, 30°C, 48 hr, with cell wall disturbing

*Figure 4 continued on next page*

*Figure 4 continued*

agents (CR, Congo Red; CFW, Calcofluor White) or drugs (nikkomycin, tunicamycin, caspofungin). These results are representative of three independent experiments. Pictures were taken with a Phenobooth (Singer Instruments). (**d**) Filamentation assay of all strains. Top row: growth in liquid SD medium at 30°C; middle panels: growth in liquid YNB medium + 2% GlcNAc, buffered at pH 7.2 at 37°C during 6 hr. Pictures were taken using an Olympus IX 83 microscope, ×40 objective. Bottom row: cells were grown on agar YNB + 2% GlcNAc, buffered at pH 7.2, at 37°C for 6 days.

The online version of this article includes the following source data and figure supplement(s) for figure 4:

**Source data 1.** Raw data for panels a and b.

**Figure supplement 1.** Control PCR control of mutants obtained in this study.

**Figure supplement 1—source data 1.** PDF file containing original gels PCR for *Figure 4—figure supplement 1c*, indicating the relevant bands.

**Figure supplement 1—source data 2.** Original files for gels PCR analysis displayed in *Figure 4—figure supplement 1c*.

**Figure supplement 2.** Control PCR of the quadruple mutant complemented for *KRE6* (kre6/kre62/skn2/skn1Δ/Δ+P_{ACT}–KRE6).

**Figure supplement 2—source data 1.** PDF file containing original gel PCR for *Figure 4—figure supplement 2*, indicating the relevant bands.

**Figure supplement 2—source data 2.** Original files for gel PCR analysis displayed in *Figure 4—figure supplement 2*.

## β-1,6-Glucan stimulates human peripheral blood mononuclear cells in vitro

The major cell wall polysaccharides of *C. albicans*, β-1,3-glucan, mannan, and chitin function as PAMPs, and their recognition by host PRRs trigger immune responses. Although β-1,6-glucan is one of the major constituent polysaccharides in the cell wall of *C. albicans*, its immunomodulatory potential has been understudied. We extracted β-1,6-glucan from the cell wall of the parental strain and used this to stimulate peripheral blood mononuclear cells (PBMCs) and neutrophils isolated from the whole blood samples of healthy human donors. As controls, PBMCs were stimulated with the AI fraction (composed of β-1,3-glucan, β-1,6-glucan, and chitin), as well as with the periodate-oxidized AI fraction (AI-OxP) devoid of β-1,6-glucan. A protein profiler was used to identify cytokines, chemokines, and acute-phase proteins differentially released by PBMCs and neutrophils stimulated with the three different cell wall fractions (AI, AI-OxP, and β-1,6-glucan) (*Figure 6—figure supplements 2 and 3*). Then, differentially stimulated cytokines, chemokines, and acute-phase proteins were quantified by ELISA. Fractions containing β-1,3-glucan (AI and AI-OxP fractions) induced the secretion of pro-inflammatory cytokines, including IL-6, IL-1β, and TNF-α, chemokines such as IL-8, MCP-1, MIP-1β, and RANTES, and the complement factor C5a (an acute-phase protein), but anti-inflammatory cytokine IL-10 was released in a relatively small amount (*Figure 6a*). However, removal of β-1,6-glucan by periodate oxidation (AI-OxP) led to a significant decrease in the IL-8, IL-6, IL-1β, TNF-α, C5a, and IL-10 released, suggesting that their stimulation was in part β-1,6-glucan dependent. Although the response was much lower than that induced by the AI fraction and AI-OxP, β-1,6-glucan activated PBMCs and induced secretion of most cytokines and chemokines (*Figure 6a*). Both AI and AI-OxP fractions stimulated neutrophils secreted IL-8 at a similar level (*Figure 6b*) but purified β-1,6-glucan did not, suggesting that β-1,6-glucan and β-1,3-glucan stimulate innate immune cells in distinct ways.

Because the structure of cell wall β-1,6-glucan is dependent on stress as well as environmental conditions of growth, we investigated the structure–function immune response of β-1,6-glucan from *C. albicans*. Shorter β-1,6-glucan chains (19 kDa) were purified from the cell wall of *C. albicans* cell wall grown with lactate as carbon source and larger β-1,6-glucan chains (70 kDa) were purified from cell wall produced in the presence of caspofungin (*Figure 2*). Both fractions stimulated PBMCs and led to cytokine/chemokine secretion at a level similar to the control β-1,6-glucan, suggesting that β-1,6-glucan size was not a significant factor influencing the stimulation of PBMCs (*Figure 6—figure supplement 1*).

## Discussion

Cell wall β-1,6-glucan is a conserved constituent in most of the major human fungal pathogens, including *Candida*, *Cryptococcus*, *Pneumocystis*, *Histoplasma*, and *Malassezia* species (*Garcia-Rubio et al., 2020*; *Stalhberger et al., 2014*; *Kottom et al., 2015*). The cell wall of fungal pathogens is the target of several classes of antifungal drugs and is critical for immune recognition and activation, and therefore has been the subject of numerous investigations (*Gow et al., 2017*; *Garcia-Rubio*

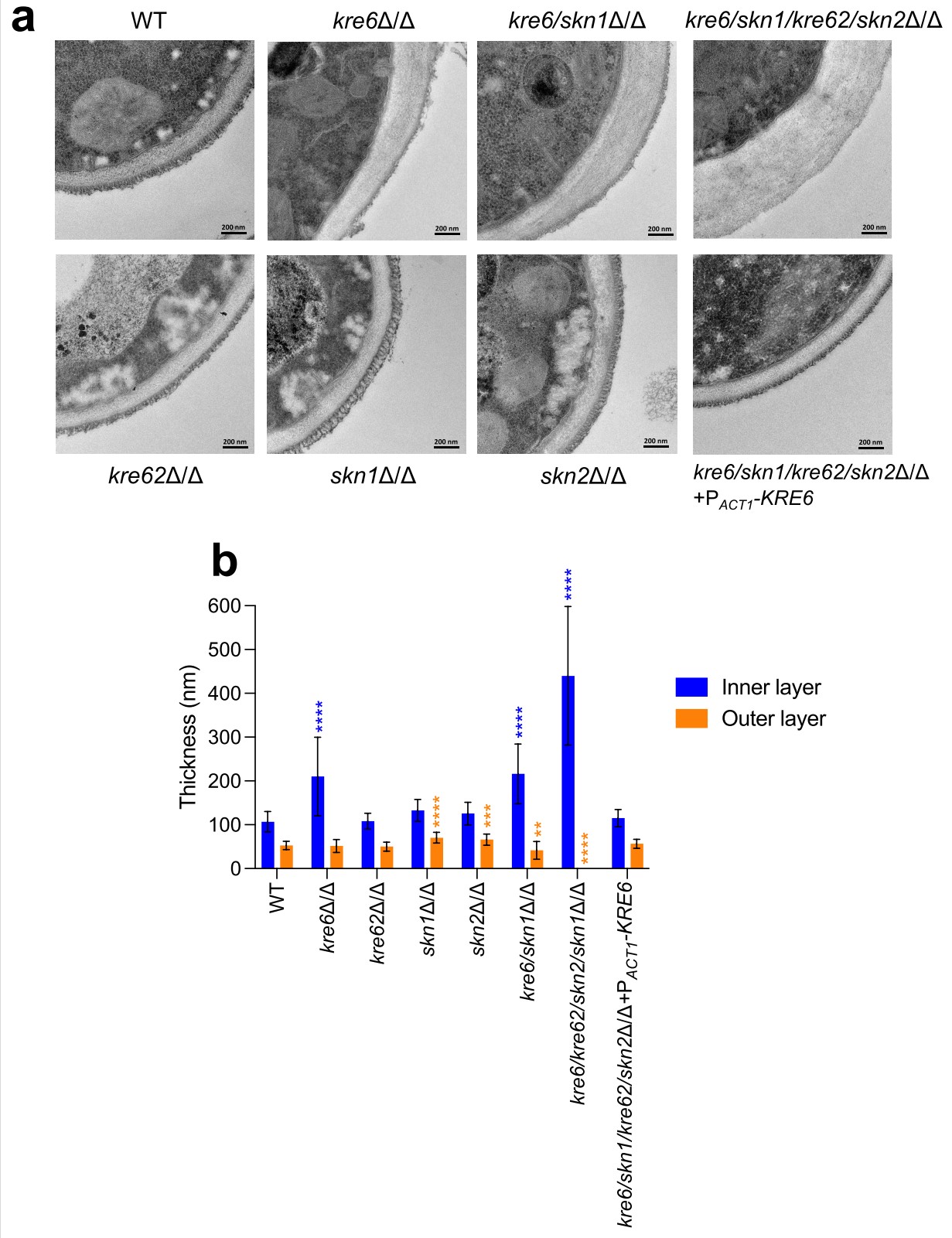

**Figure 5.** Cell wall electron microscopy observations of *KRE6* simple and multiple mutants. (**a**) Representative transmission electron microscopy images of the cell wall of each strains. After culture, cells were fixed and high-pressure frozen and freeze substituted with Spurr resin. Sections were cut and stained and then pictures were taken with a Tecnai Spirit 120Kv TEM microscope. Scale bar = 200 nm. (**b**) Measurement of the inner and outer cell wall

*Figure 5 continued on next page*

*Figure 5 continued*

layers of the mutants. Means and standard deviations are represented. 37–40 measurements were performed randomly on 7–13 cells. Statistical analyses were performed with one-way ANOVA with Tukey's multiple comparisons test: *$p<0.05$; **$p<0.01$; ***$p<0.001$; ****$p<0.0001$.

The online version of this article includes the following source data for figure 5:

**Source data 1.** Raw data for panel b.

---

*et al., 2020*). However, this polysaccharide has been neglected, perhaps because its analysis requires bespoke methodologies. Here, we established new biochemical tools for quantitative and comparative analysis of cell wall β-1,6-glucan dynamics and structure in *C. albicans* and used them to demonstrate that this polysaccharide is instrumental in determining the gross architecture of the fungal wall and its immune reactivity.

In SD medium at 37°C, β-1,6-glucan represents 23% of the total cell wall, with an average size of 58 kDa (*Figure 1*). From our analytical data on *C. albicans* cell wall, we proposed an updated model underlining the true role of β-1,6-glucan as a major cell wall component (*Figure 7a*). NMR has demonstrated that the structure of β-1,6-glucan in the yeast cell wall is similar between species (*Aimanianda et al., 2009*; *Stalhberger et al., 2014*), but there are variations in β-1,6-glucan content, chain length, and branching. *Malassezia restricta* contains long β-1,6-glucan chains, averaging 80 kDa, whilst in *S. cerevisiae* and *C. albicans* they are moderate in size (30–60 kDa). We showed here that modulating environmental conditions induce significant changes in both cell wall structure and content. As described previously, lactate used as the sole carbon source led to a thinner inner cell wall layer and greater exposure of β-1,3-glucan (*Ene et al., 2012*; *Ballou et al., 2016*). We observe here that lactate medium also led to a marked reduction in cell wall β-1,6-glucan content and chain length (*Figure 2*, *Figure 7b*). Our data showed that several factors related to host niches (morphology, temperature, salinity, pH, and presence of organic acids) and the presence of antifungals had a significant impact on the structure, size, and branching of β-1,6-glucan (*Figure 7b*). This dynamic nature of the β-1,6-glucan component of the wall (*Figure 2*, *Figure 2—figure supplement 1*, *Figure 3—figure supplement 1*) is as important as the regulation and structural diversity of chitin, mannan, and β-1,3-glucan (*Gow and Lenardon, 2023*).

The dynamic regulation of mannan biosynthesis leading to structural changes in the outer mannoprotein layer of *C. albicans* is also under-investigated. *C. albicans* evolves in different environments as a commensal or pathogen (*d'Enfert et al., 2021*), mainly in the gut, oral cavity, genital tract, and blood. How fungal cell walls adapt to changing environments was recently been listed as one of the top five unanswered question in fungal cell surface research (*Gow et al., 2023*). Our data demonstrate that in response to several environmental factors β-1,6-glucan and mannan productions are coupled. The in vivo importance of this novel dynamic process remains to be addressed but it could be critical in the articulation of host inflammatory responses (*de Assis et al., 2022*; *Chen et al., 2022*).

Immunolabeling with an anti-β-1,6-glucan serum demonstrated that this polymer is exposed at the cell surface and is therefore accessible to receptors that stimulate human immune cells (*Figure 6—figure supplement 5*, *Figure 6*). Similarly in *Pneumocystis carinii*, β-1,6-glucans are exposed to the cell surface and stimulate mouse macrophages, leading to the release of TNF-α (*Kottom et al., 2015*). Our immunological data indicate that β-1,6-glucan is a bona fide fungal cell wall PAMP that activates the human immune system (*Figure 7d*). Beads coated with pustulan (an insoluble linear form of β-1,6-glucan) activate neutrophils and induce massive ROS production and phagocytosis in a dectin-1-independent manner (*Rubin-Bejerano et al., 2007*; *Palma et al., 2006*). We observed that the absence of human serum in the culture medium led to a strong reduction in β-1,6-glucan-induced cytokine secretion from PBMCs and that β-1,6-glucan bind to C3b (*Figure 6—figure supplement 4*). These data are in agreement with the binding of pustulan to C3b/C3d, as described previously (*Rubin-Bejerano et al., 2007*; *Palma et al., 2006*). This allows us to assume that the opsonization by complement component C3 could partially mediate the recognition of β-1,6-glucan by immune cells, but needs to be validated experimentally. We observed that the size of soluble β-1,6-glucan did not alter immune stimulation in vitro (*Figure 6—figure supplement 1*), whereas, despite its ability to bind soluble and particulate β-1,3-glucan polymers, Dectin-1 signaling is only activated by particulate β-1,3-glucans (*Goodridge et al., 2011*). The relevance of fungal β-1,6-glucan to the global host immune response and mechanism of recognition remains to be investigated.

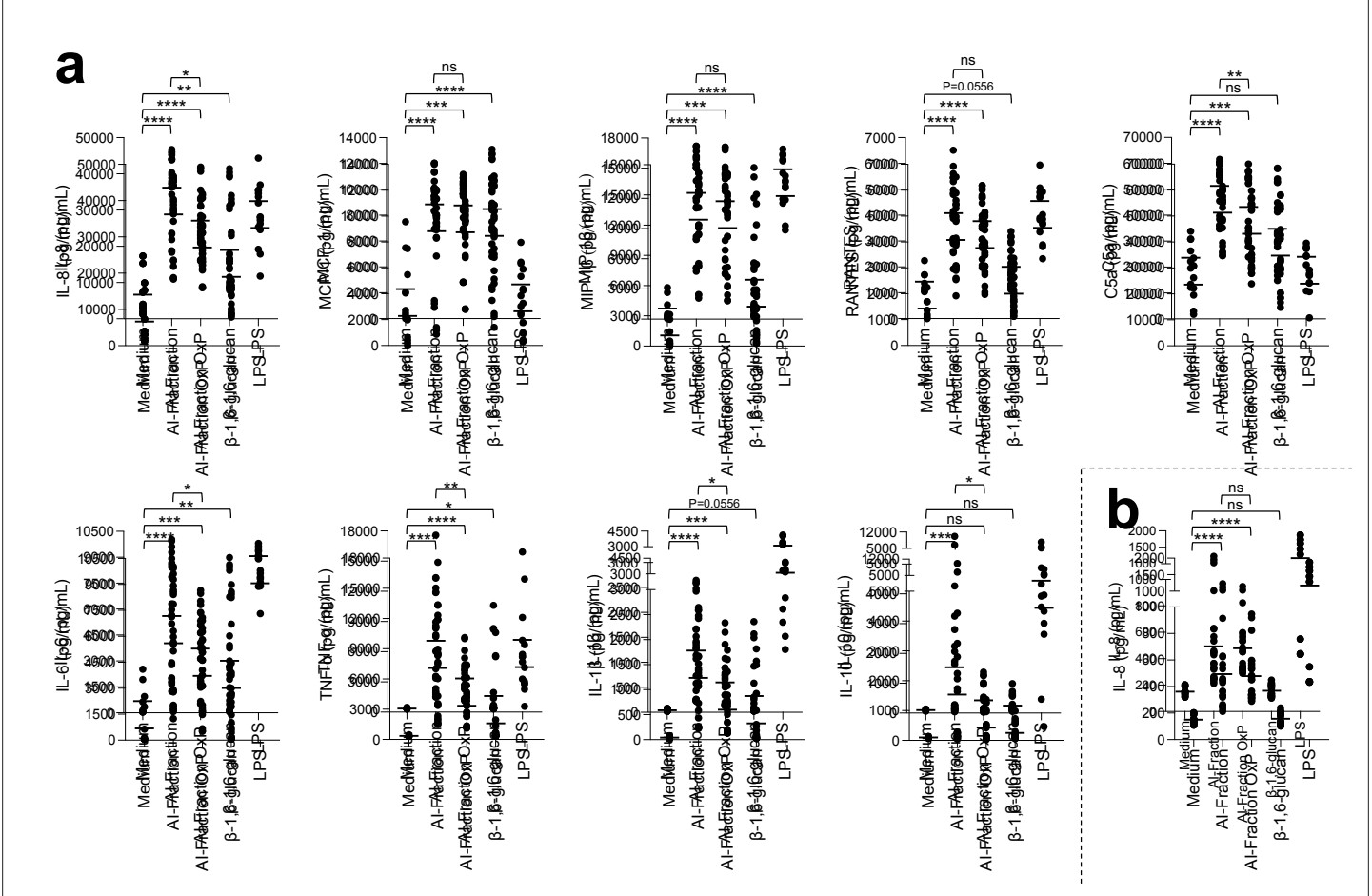

**Figure 6.** Stimulation of peripheral blood mononuclear cells (PBMCs) and neutrophils in vitro by cell wall fractions and purified β-1,6-glucans from *C. albicans*. Cytokines, chemokines, or acute-phase proteins (IL-8, MCP-1, IL-6, MIP-1β, IL-1β, TNF-α, RANTES, C5a, IL-10) concentrations in culture supernatants of PBMCs (**a**) and neutrophils (**b**) stimulated by cell wall fractions of *C. albicans* (AI-Fraction, AI-Fraction OxP, and β-1,6-glucan) at 25 µg/mL or LPS (positive control, 0.1 µg/mL). PBMCs and neutrophils were isolated from healthy human donors (n = 8). Three independent batches of each fractions were used. Means are represented and data were analyzed using nonparametric Friedman test with Dunn's multiple comparisons: *p<0.05; **p<0.01; ***p<0.001; ****p<0.0001; ns, nonsignificant.

The online version of this article includes the following source data and figure supplement(s) for figure 6:

**Source data 1.** Raw data of panels a and b.

**Figure supplement 1.** Stimulation of peripheral blood mononuclear cells (PBMCs) and neutrophils in vitro by β-1,6-glucan with different size from *C. albicans*.

**Figure supplement 1—source data 1.** Raw data of panels a and b.

**Figure supplement 2.** Human proteome profiler done with culture supernatant from peripheral blood mononuclear cells (PBMCs) stimulated with *C. albicans* cell wall fractions.

**Figure supplement 2—source data 1.** PDF file containing original blots for *Figure 6—figure supplement 2*, indicating the relevant bands.

**Figure supplement 2—source data 2.** Original files for blots analysis displayed in *Figure 6—figure supplement 2*.

**Figure supplement 3.** Human proteome profiler done with culture supernatant from neutrophils stimulated with *C. albicans* cell wall fractions.

**Figure supplement 3—source data 1.** PDF file containing original blots for *Figure 6—figure supplement 3*, indicating the relevant bands.

**Figure supplement 3—source data 2.** Original files for blots analysis displayed in *Figure 6—figure supplement 3*.

**Figure supplement 4.** β-1,6-Glucan from *C. albicans* activates complement system.

**Figure supplement 4—source data 1.** Raw data of panels a and b.

**Figure supplement 5.** Exposure of β-1,6-glucans and β-1,3-glucans at the cell surface of *C. albicans* SC5314.

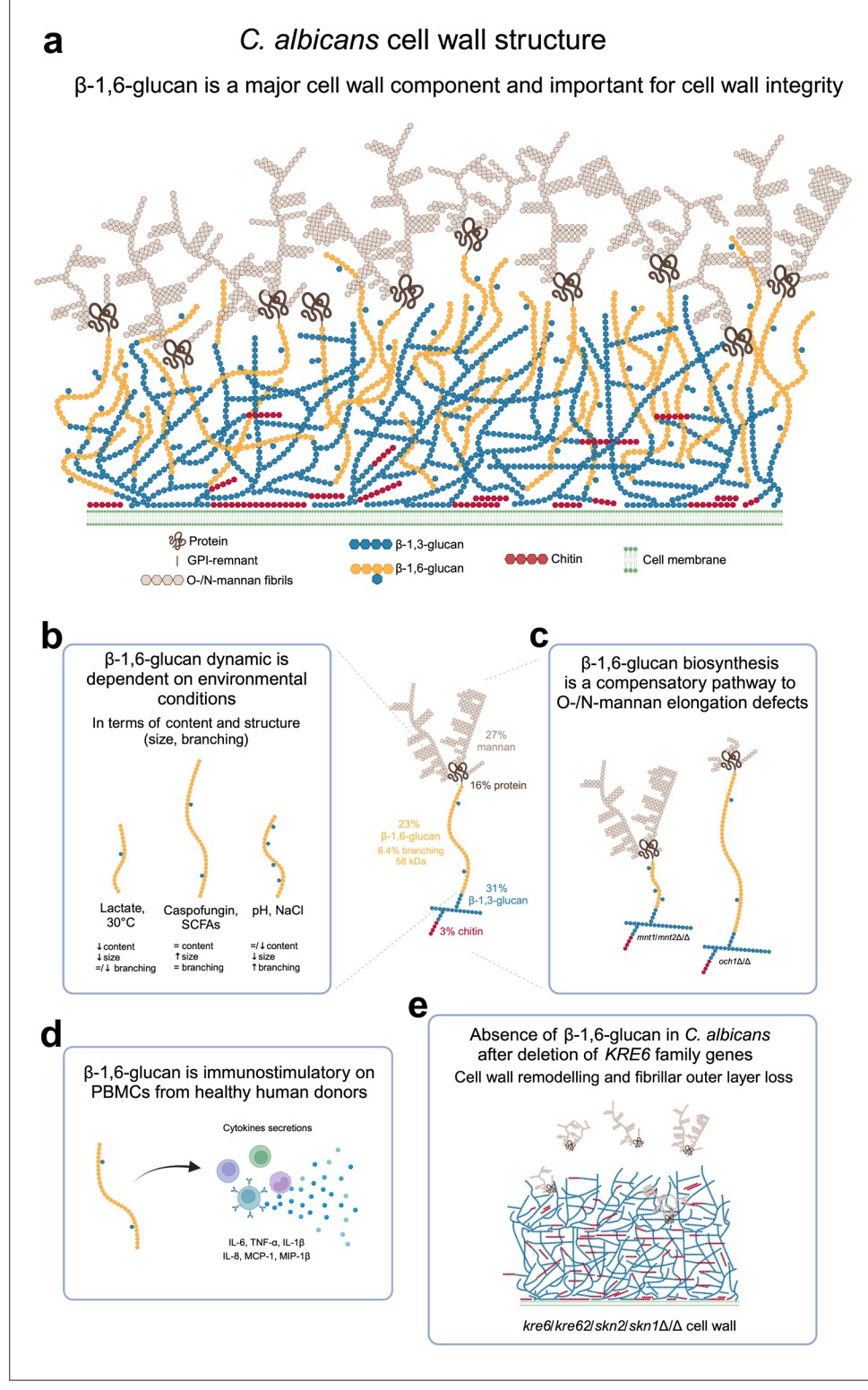

**Figure 7.** β-1,6-Glucan in *C. albicans* is a major and dynamic cell wall polymer. (**a**) Scheme of the cell wall of *C. albicans*. The proportion of each cell wall polymer was representative of the results obtained on *C. albicans* SC5314 grown in liquid synthetic dextrose (SD) medium at 37°C. (**b**) Scheme representing the dynamic of β-1,6-glucan under different environmental factors. (**c**) β-1,6-Glucan is a compensatory pathway for mannan elongation defect. (**d**) β-1,6-Glucan is a PAMP. (**e**) Scheme of the cell wall of *KRE6* family deficient mutant. Created with BioRender. com.

*Figure 7 continued on next page*

*Figure 7 continued*

The online version of this article includes the following figure supplement(s) for figure 7:

**Figure supplement 1.** A model for β-1,6-glucan biosynthetic pathway and putative role of Kre6 family members in this process in yeast.

In *C. albicans*, β-1,3-glucan exposure is regulated by external signals such as hypoxia, carbon source, or iron deprivation via the upregulation of β-1,3-glucanase activities that lead to β-1,3-glucan masking and immune evasion (*de Assis et al., 2022*; *Childers et al., 2020b*; *Yang et al., 2022*). Regulation of the mechanisms determining β-1,6-glucan exposure and the expression of secreted glucanases remain also unknown in *C. albicans*. The absence of β-1,6-glucan in the AI fraction led to a reduced inflammatory response (*Figure 6*), suggesting that an endo-β-1,3-glucanase, such as Eng1, may enable the release of cross-linked β-1,6-glucan from the cell surface, which could attenuate the host response by pruning β-1,6-glucans that are covalently bound to β-1,3-glucans from the surface (*Yang et al., 2022*).

Cell wall organization in *C. albicans* depends on the cross-linking of all polymers that function as a dynamic flexible network (*Gow et al., 2017*; *Garcia-Rubio et al., 2020*). Their interconnectedness means that alteration in the synthesis of one polymer can lead to a compensatory change and cell wall remodeling. Our data provides one new example of this phenomenon, namely decrease in cell wall chitin content results in an increase in both β-1,3-glucans and β-1,6-glucans in the AS fraction of the wall. Chitin exists in the yeast cell wall in three forms: free (~40%), and cross-linked to β-1,3-glucan (40–45%) or β-1,6-glucan (15–20%) and cross-linked β-glucan-chitin forms the core fibrillar cell wall structure resistant to alkali solubilization (*Gow et al., 2017*; *Cabib, 2009*; *Cabib and Durán, 2005*; *Cabib et al., 2007*). Therefore, a decrease in the chitin content might reduce the formation of fibrillar structure, thereby releasing the β-glucan contents into the AS fraction. On the other hand, in the *FKS1* mutant or upon caspofungin treatment, cell wall chitin and β-glucan-chitin contents were increased, which could be attributed to an upregulated *CRH1* expression due to decreased β-1,3-glucan synthesis as in *FKS1* mutant (*Terashima et al., 2000*) or due to caspofungin treatment. *CRH* family members are involved in the β-glucan-chitin cross-linking (*Cabib, 2009*). Interestingly, defects in the expression of *CHS1* that encodes an essential chitin synthase required to construct the primary septum were also compensated by an increased content of cell wall β-1,6-glucan but not of β-1,3-glucan (*Figure 3*). This suggests that Chs1 has other functions beyond septum construction that had previously been suggested by loss of cell integrity and swelling of cells upon *CHS1* repression (*Munro et al., 2001*). In *S. cerevisiae*, deletion of *GAS1*, the ortholog of *PHR1* and *PHR2*, led to a strong decrease in cross-linked β-1,6-glucan to β-1,3-glucan-chitin cell wall core (*Aimanianda et al., 2017*). Deletions of *PHR1* and *PHR2*, holding a β-1,3-glucan remodeling activity, led to an increased chitin content (*Figure 3—figure supplement 1*) and a reduction in β-1,3-glucan branching (*Figure 3—figure supplement 2*). However, a decrease in β-1,6-glucan content and β-1,6-glucan size was only observed in the *PHR2* mutant (*Figure 3*). As *PHR1* and *PHR2* genes are strongly regulated by external pH, the compensatory differences we described may be explained by pH-dependent regulation of β-1,6-glucan synthesis (*Figure 3*).

In *C. albicans*, the mannan moieties of glycoproteins are covalently cross-linked to the β-glucan-chitin core. Mutants defective in *N*-mannan elongation led to a marked increase in β-1,6-glucan content and size (*Figure 3*). In yeast, Och1 and Mnn9 are α-1,6-mannosyltransferases that are involved in initial steps of α-1,6-mannan polymerization in the Golgi. Members of the Mnn2 α-1,2 mannosyltransferases are required for the addition of the side chains of the α-1,6-mannan backbone (*Hall and Gow, 2013*). Mutants in these genes led to defects in the outer fibrillar layer (*Lenardon et al., 2020*; *Southard et al., 1999*) and increased β-1,3-glucan exposure (*Hall and Gow, 2013*; *Yadav et al., 2020*). The quantification of β-1,6-glucan content in these *N*-mannan-deficient mutants had not been described previously in *C. albicans* (*Hall and Gow, 2013*; *Southard et al., 1999*) and underline the new finding that the regulation of mannan, β-glucans, and chitin synthesis is coordinated and coupled. Screening with K1 toxin killer revealed a hypersensitivity of the Sc*mnn9* mutant and an increase in β-1,6-glucan cell wall content (*Pagé et al., 2003*), reinforcing the hypothesis that β-1,6-glucans play a compensatory role when *N*-mannosylation is compromised (*Figure 7c*). Thus, a defect in the fibrillar outer mannan layer led to an increase in the inner cell wall chitin and β-1,6-glucan contents (*Figure 3—figure supplement 1*), suggesting that their synthesis was regulated by a rescue mechanism associated

with the reprogramming of specific transcriptional responses via the cell wall integrity (CWI) pathway (**Childers et al., 2020a**). Defects in *O*-mannosylation also affected β-1,6-glucan synthesis, leading to shortened highly branched β-1,6-glucan chains, without significant changes in overall cell wall composition, suggesting a specific inhibition of the elongation of β-1,6-glucan chain and showing that size and branching of β-1,6-glucan can be regulated independently (**Figure 7c**).

Fine details of the biosynthetic pathway of yeast β-1,6-glucans are not clear. In *S. cerevisiae*, deletion of *KRE5* and *ROT2* led to a reduction in cell wall β-1,6-glucans (**Shahinian and Bussey, 2000**). In *C. albicans*, this reduction is characterized by the presence of short seldom-branched β-1,6-glucan chains (**Figure 3**). Although no difference was observed in the *CWH41* mutant, our data imply that canonical *N*-glycan processing in the ER is required for complete β-1,6-glucan synthesis (**Pagé et al., 2003**), and that Kre5p is not the β-1,6-glucan polymerase (**Herrero et al., 2004**). Furthermore, our results confirm that the β-1,6-glucan biosynthetic pathway is similar in these phylogenetically distant yeasts, and that the *N*-glycosylation pathway is critical for β-1,6-glucan synthesis. Defects in early *N*-glycan synthesis at the ER level led to concomitant defects in β-1,6-glucan synthesis while defects in *N*-mannan elongation led to increased β-1,6-glucan synthesis. However, as Kre5, Rot2, and Cwh41 are part of the calnexin cycle involved in the control of N-glycoprotein folding in the ER, early N-glycan synthesis may have an indirect effect in ensuring the functionality of the enzymes rather than the synthesis of N-glycan moiety required for β-1,6-glucan synthesis. Our hypothesis is that this biosynthesis begins intracellularly. According to in vitro systems in *S. cerevisiae*, the initial step is the polymerization of a linear chain by a β-glucosyltransferase using UDP-glucose as the sugar donor (**Aimanianda et al., 2009**; **Vink et al., 2004**) and probably an as yet unknown acceptor (**Figure 7—figure supplement 1**). The second step would be the addition of glucoside and laminaribioside as a side chain. Finally, after release into the cell wall, β-1,6-glucan will be cross-linked to the other cell wall polymers. Among the *KRE* genes, only *KRE6* may be directly involved in β-1,6-glucan synthesis. Kre6 and its homologs contain a β-glycosyl-hydrolase domain of the GH-16 family, which includes a large number of fungal enzymes such as Crh1 homologs that are involved in the cross-linking of chitin and β-1,3-glucan (**Cabib, 2009**). In *C. albicans*, the quadruple deletion of *KRE6*, *SKN1*, *SKN2*, and *KRE62* led to the complete absence of β-1,6-glucan in the cell wall (**Figures 3 and 7e**) but also in culture supernatants and in intracellular extracts (not shown), demonstrating a stop of β-1,6-glucan synthesis rather than miss-localization. The absence of β-1,6-glucan in the quadruple mutant led to global cell wall remodeling and the absence of the outer fibrillar *N*-mannan layer (**Figures 5 and 7e**). Kre6 had the dominant role and its absence led to a hypersensitivity to cell wall-targeting antifungal drugs (**Mio et al., 1997**; **Han et al., 2019b**). One exception was observed for caspofungin, which enhanced the growth of all *KRE6* deleted mutants (**Figure 4c**). Growth above the MIC (paradoxical growth) has been described at high caspofungin concentrations and is associated with high chitin content (**Han et al., 2019a**; **Rueda et al., 2014**) as observed in these mutants. Our data show that Kre6 is required for the production of normal amounts of β-1,6-glucan and that the remaining β-1,6-glucans present in *kre6Δ/Δ* was composed of short and highly branched chains. This suggests that the Kre6 family is required for the polymerization and/or elongation of β-1,6-glucan, but not glucan branching.

In *S. cerevisiae*, only two *KRE6* family members have been identified, Kre6 and Skn1, and their absence led to strong growth defects and hypersensitivity to cell wall perturbing agents – similar to what we observed here in *C. albicans* (**Roemer et al., 1993**). In *C. neoformans*, among the six *KRE6* homologs, *KRE6* and *SKN1* are required for the production of cell wall β-1,6-glucan (**Gilbert et al., 2010**), and although the *KRE6* family is essential for β-1,6-glucan synthesis, its biochemical function remains unknown. Kre6 homologs are type II membrane proteins with a GH-16 domain at the luminal C-terminus and an unfolded cytosolic *N*-terminal tail (**Kurita et al., 2011**). Yeast Kre6 and Skn1 proteins have moderate similarity with diatom TGS (1,6-β-transglycosylase) proteins (**Kroth et al., 2008**; **Huang et al., 2016**) and can synthesize glucan from UDP-Glc (**Inukai et al., 2023**). Kre6 homologs share structural similarities with β-glucanases and β-transglycosylases, arguing for a possible cell wall remodeling function. In *S. cerevisiae*, Kre6p has been localized in the endoplasmic reticulum, the plasma membrane, and secretory vesicle-like compartments (**Shahinian and Bussey, 2000**; **Kurita et al., 2011**; **Kurita et al., 2012**), and the cytosol tail was shown to be essential for the correct cellular localization and β-1,6-glucan synthesis (**Kurita et al., 2011**). Surprisingly, in *C. albicans* the cytosol tail is not essential (**Han et al., 2019b**), suggesting some functional differences of this protein in these yeasts. In *C. albicans*, the double *KRE6/SKN1* mutant is avirulent in a mice model of candidiasis

(*Han et al., 2019b*), showing that the β-1,6-glucan biosynthetic pathway is a potential virulence target. However, the drugs jervine (steroidal alkaloid) and a pyridobenzimidazole derivate (D75-4590), which may target Kre6 and inhibit *S. cerevisiae* growth (*Kubo et al., 2022*; *Kitamura et al., 2009*), had no effect on *C. albicans* (*Kubo et al., 2022*; *Kitamura et al., 2009*), presumably representing species-specific differences in drug profiles. Kre6 family members are present in all fungal pathogens producing cell wall β-1,6-glucans, and further investigations are required to understand the enzyme activities and the diversity of structures related to the potential drug targeting of these proteins.

Overall, this work highlights the central role that β-1,6-glucan plays in orchestrating cell wall synthesis during growth and in response to stress and sets the focus for further work that can exploit this central role in mitigating fungal infections.

## Materials and methods
### Strains and growth conditions
The *C. albicans* strains used in this study are listed in *Supplementary file 1*. The reference strain SC5314, used for all stress conditions, was grown overnight in 25 mL flasks, at 37°C, 180 rpm, in SD medium (2% glucose, 0.67% yeast nitrogen base with amino acids [YNB] [BD], pH 5.4). Overnight cultures were diluted to an $OD_{600}$ = 0.001, in 50 mL fresh SD medium, incubated at 37°C, 180 rpm, and then harvested either when reaching mid-exponential phase ($OD_{600}$ 3–5) or at stationary phase ($OD_{600}$ 11–12 after 30 hr). The following growth conditions were selected: hypha induction (SD containing 2% GlcNAc instead of 2% glucose, pH 7.2, no yeast form was observed in this medium), temperature (30°C), osmolarity (1 M NaCl), carbon source (SD containing 2% lactate [ChemCruz] instead of 2% glucose), oxygen limitation (hypoxia: 1% $O_2$, 5% $CO_2$, anaerobic chamber [Baker Ruskinn, I-CONIC, Bugbox M]), and oxidative stress ($H_2O_2$ 1.5 mM, Prolabo). For studies of the effect of medium pH, SD medium was buffered with 100 mM MES (Sigma) at pH 4 or 5.4 or with MOPS (Sigma) at pH 7.5. For experiments on the effects of short-chain fatty acids (SCFAs), SD medium was supplemented with 100 mM MES and 50 mM acetate (Sigma) or 20 mM propionate (Sigma) or 15 mM butyrate (Sigma) and buffered to pH 5.4. Drugs were tested at sublethal concentrations: Calcofluor White (CFW, 400 µg/mL, Sigma-Aldrich), Congo Red (CR, 50 µg/mL, Sigma), tunicamycin (0.125 µg/mL, Sigma), nikkomycin (16 µg/mL, Sigma), and caspofungin (0.015 µg/mL, Sigma). Biofilm production was induced in SD medium at 37°C for 48 hr in six-well plates as previously described with some modifications (*Rai et al., 2023*). Overnight cultures were diluted at $OD_{600}$ = 0.2 in six-well polystyrene plates (TPP) in 2 mL of minimal medium (SD: 2% glucose, 0.67% yeast nitrogen base without amino acids [BD], pH 5.4) supplemented by arginine 0.1 g/L, histidine 0.1 g/L, uridine 0.02 g/L, and methionine 0.2 g/L. The plates were incubated at 37°C, 60 min at 110 rpm for initial cell adhesion. Then, the plates were washed with 2 mL of fresh medium, and 4 mL of fresh medium was added. Plates were then sealed with Breathseal sealing membranes (Greiner Bio-one) and incubated at 37°C, 110 rpm. After 48 hr, supernatant culture was aspirated, the wells were gently washed with PBS, and biofilm was collected in PBS.

Cell wall mutant strains were grown at 30°C, 180 rpm, in SD medium supplemented with arginine (100 µg/mL), uridine (100 µg/mL), and histidine (20 µg/mL). The *phr1Δ/Δ* mutant was grown in SD buffered medium at pH 7.5 (with 100 mM MOPS) and the *phr2Δ/Δ* mutant at pH 5.5 (with 100 mM MES). The conditional mutant $P_{MRP1}$-*CHS1/chs1Δ* was first precultured in YNB with 2% maltose as a carbon source to allow the *MRP1* promoter expression and then it was grown, as all other mutants, in SD (carbon source glucose), under which condition the promoter is repressed.

### Cell wall fractionation
After growth culture, the fungal biomass was collected by centrifugation (5 min, 3300 × *g*), washed, and disrupted in 200 mM Tris-HCl pH 8 using a cell disruptor (FastPrep, MPbio, six cycles, 6 m/s, 60 s) with 0.3 mm glass beads (Braun). The cell wall was collected by centrifugation (5 min, 3300 × *g*), washed twice with 200 mM Tris-HCl, pH 8, then twice with milliQ water and freeze-dried. The wall polymers were then fractionated as previously described (*Liu et al., 2023*). Briefly, noncovalently bound proteins were extracted by the following treatment: dried cell wall was treated twice in 50 mM Tris-HCl, 50 mM EDTA, pH 7.5, 2% SDS, 40 mM β-mercaptoethanol in boiling water for 10 min (10–20 mg of cell wall/mL). After centrifugation (10 min, 3300 × *g*), three volumes of ethanol

were added to the supernatant and kept overnight at +4°C. The precipitate (SDS-β-ME fraction) was collected by centrifugation (10 min, 3300 × $g$) and dialyzed against water before freeze-drying. Other cell wall pellet was washed three times with 150 mM NaCl and extracted twice with a 1 M NaOH containing 1 mg/mL BH$_4$Na (Sigma), at 10–20 mg of cell wall/mL at 65°C for 1 hr. After centrifugation (10 min, 3300 × $g$), the alkali-soluble extract (AS fraction, supernatant) was neutralized with acetic acid, extensively dialyzed against water, and freeze-dried. The alkali-insoluble fraction (AI fraction, pellet) was washed three times with distilled water and freeze-dried. Of note, the cell disruption was a key step to remove glycogen from cell wall extracts (*Figure 1—figure supplement 2*).

## Cell wall polymers quantification

Cell wall polymers of the three cell wall fractions (SDS-β-ME, AI, and AS fractions) were quantified as previously described (*Liu et al., 2023*). Briefly, total neutral sugars were quantified by the phenol-sulfuric acid colorimetric assay using glucose as a standard (*DuBois et al., 1956*). Proteins were quantified by colorimetry using the Pierce BCA Protein Assay Kit (Thermo Scientific) according to the manufacturer's instructions. Monosaccharides, glucose and mannose, were identified and quantified by gas liquid chromatography (Perichrom) as their alditol acetates obtained after hydrolysis (4 N trifluoroacetic acid [TFA, Sigma], 100°C, 4 hr) (*Sawardeker et al., 1965*). Glucosamine was quantified by HPAEC (Dionex, ICS-3000) on a CarboPAC-PA1 column (Thermo Scientific, 250 mm × 4.6 mm) after acid hydrolysis (6 M HCl, 6 hr, 100°C) using glucosamine as a standard. Samples were eluted at a flow rate of 1 mL/min with 18 mM NaOH for 15 min, then with a linear gradient to 300 mM sodium acetate (AcONa) in 100 mM NaOH for 20 min and finally under isocratic conditions with 300 mM AcONa in 100 mM NaOH for 5 min. The column was equilibrated for 20 min with the initial buffer before injection. Samples were detected on a pulsed electrochemical detector.

## Quantification of β-1,6-glucan in AI and AS fractions

To quantify β-1,6-glucan in the AI fraction, a colorimetric assay was used. Oxidation with periodate allows specific degradation of vicinal carbons bearing hydroxyl groups. In the AI fraction, only β-1,6-glucans were sensitive to this treatment, leading to the formation of aldehyde functions, which were quantified in the presence of 4-hydroxybenzhydrazide (PAHBAH) (Sigma) (*Figure 1—figure supplement 3*). Oxidation was carried out at 50°C for 6 hr on 100 μL of AI fraction (0.5 mg/mL) with 100 μL of 25 mM $m$-IO$_4$Na (Sigma-Aldrich). Excess of reagent was quenched with 100 μL sodium sulfite 200 mM (Sigma). Then, 100 μL of reaction mixture were added to 900 μL of PAHBAH reagent (0.63 g sodium sulfite, 5 mL NaOH 5 M, 5 mL sodium citrate 0.5 M, 5 mL NaCl$_2$ 0.2 M qsp 100 mL milliQ water, 1 g PAHBAH) and placed in a boiling water bath for 10 min. The quantification of produced aldehyde was measured by the absorbance at 405 nm. Pustulan (linear β-1,6-glucan, Elicityl OligoTech) was used as a standard. The β-1,3-glucan content of the AI fraction was calculated as the total glucose content after subtraction of the amount of β-1,6-glucan.

The presence of mannan in the AS fraction prevents use of the colorimetric assay of β-1,6-glucan after periodate oxidation. β-1,3-Glucan and β-1,6-glucan of the AS fraction were therefore quantified by enzyme digestion. The AS fraction was digested with the endo-β-1,3-glucanase, LamA (from *Thermotoga neapolitana* expressed in *Escherichia coli*) (*Zverlov et al., 1997*) or with an endo-β-1,6-glucanase (from *Schizosaccharomyces pombe* expressed in *Pichia pastoris*) (*Dueñas-Santero et al., 2010*). Digestions were carried out at 37°C during 24 hr by treating 0.2 mg of AS fraction with 20 μL of LamA (specific activity: 10 μmol eq/min/μL) or 60 μL of the endo-β-1,6-glucanase (specific activity: $3.03 \times 10^{-6}$ μmol eq/min/μL), in 50 mM NaOAc, pH 6, in a final volume of 200 μL. Both enzymatic treatments released more than 95% of glucan. Products were quantified by the amount of released reducing sugars using the PAHBAH reagent (*Fontaine et al., 1997*) and identified by HPAEC on a CarboPAC-PA1 column (Thermo Scientific). Soluble and purified β-1,6-glucans from *C. albicans* and a commercial source of laminarin (Sigma) were used as standards. Products from enzymatic digestion were eluted under the following gradient: isocratic step of 98% of eluent A (50 mM NaOH) and 2% of eluent B (500 mM AcONa in 50 mM NaOH) for 2 min, 2–15 min of a linear gradient of AcONa in 50 mM NaOH (2% B to 20% B), 15–20 min of a linear gradient (20% B to 43% B), 20–22 min of a linear gradient (43% B to 100% B), 22–25 min under isocratic conditions with 100% B. The column was equilibrated for 20 min with initial buffer before injection.

## Structural characterization of β-1,6-glucan

β-1,6-Glucan are branched by β-1,3-linked glucose or laminaribiose on the main chain (*Iorio et al., 2008*; *Aimanianda et al., 2009*). Branching was estimated by enzymatic digestion by an endo-β-1,6-glucanase as previously described (*Aimanianda et al., 2009*). A 0.1 mg sample of the AI fraction was digested for 48 hr at 37°C in a final volume of 100 µL with 30 µL of endo-β-1,6-glucanase in 50 mM sodium acetate, pH 6, 5 mM $NaN_3$. Degradation products were analyzed by HPAEC on a CarboPAC-PA1 column as described above. Branching was estimated by the ratio of branched oligosaccharides on total degradation products, to which a coefficient of correction was applied. This coefficient (43.1) was calculated from a branching rate of 6.9% of purified β-1,6-glucan established by NMR (see below and *Figure 1—figure supplement 1*). Branching of β-1,3-glucan was determined in the same way after digestion of the AI fraction with LamA.

The size of the β-1,6-glucan chain was estimated by gel filtration chromatography. First, β-1,6-glucans were released by incubation at 37°C for 48 hr of the AI fraction (0.5 mg) with LamA (100 µL) in 70 mM sodium acetate, pH 6, 5 mM $NaN_3$, in a final volume of 700 µL. After 3 min centrifugation at $12,000 \times g$, the supernatant was concentrated under vacuum (SpeedVac concentrator). Then, the sample was submitted to a gel filtration on a Superdex 200 column (300 × 10 mm, Cytiva), eluted with ammonium acetate 150 mM, pH 4 at a flow rate of 0.2 mL/min. Samples were detected by a refractive index detector (Iota-2, Precision Instruments). The column was calibrated with dextran molecular weight standards (6, 10, 40, 70, and 500 kDa; Sigma-Aldrich, Pharmacia Fine Chemicals).

## Nuclear magnetic resonance (NMR)

NMR experiments were recorded at 318.15 K using a Bruker (Billerica, USA) 800 MHz Avance NEO spectrometer with an 18.8 Tesla magnetic field equipped with a cryogenically cooled triple resonance ($^1$H,$^{13}$C,$^{15}$N) TCI probe. Spectra were recorded using TopSpin 4.2 (Bruker) and analyzed with CCPNMR Analysis 2.5.2 (*Vranken et al., 2005*). $^1$H and $^{13}$C chemical shifts were referenced to external DSS (2,2-dimethyl-2-silapentane-5-sulfonate, sodium salt). After three cycles of exchange (dissolution/lyophilization) against $D_2O$, 5 mg of purified polysaccharide were dissolved in 550 µL of $D_2O$ (99.99% 2 H, Eurisotop, Saclay, France) and placed in a 5 mm tube (Norell HT). Resonance assignment, glycosidic bonds identification, and J coupling determination were achieved following standard procedures from homonuclear $^1$H 1D, 2D COSY (*Ancian et al., 1997*), NOESY (*Thrippleton and Keeler, 2003*) (150 ms mixing time), and natural abundance heteronuclear $^1$H-$^{13}$C experiments: edited HSQC (*Willker et al., 1993*; *Kay et al., 1992*), CLIP-HSQC (*Enthart et al., 2008*), H2BC (*Nyberg et al., 2005*), HMBC (*Cicero et al., 2001*), and HSQC-TOCSY (*Kay et al., 1992*) (100 ms mixing time). The anomeric configuration of monosaccharide residues was established from the corresponding chemical shifts and $^1J_{H_1\text{-}C_1}$ coupling constants obtained from CLIP-HSQC spectra. Glycosidic bonds were identified using HMBC experiments and confirmed with NOESY data. $^3J_{H_1\text{-}H_2}$ coupling constants were measured from $^1$H 1D spectra. The relative amount of monosaccharide residues was estimated from the integrals on $^1$H 1D spectra obtained at 333.15 K using a total recovery delay of 12 s.

## Transmission electron microscopy

Cells were grown as described above, and volumes of culture media equivalent to a quantity of cells equivalent to $OD_{600nm} = 1$ were collected and centrifuged 5 min at $3300 \times g$. Cells were chemically fixed at room temperature in the dark for 1 hr with 2.5% glutaraldehyde (Sigma) in fresh culture media (10 mL), washed in 1× PHEM buffer (60 mM PIPES, 25 mM HEPES, 10 mM EGTA, 2 mM $MgCl_2$, pH 7.3), and centrifugated at low speed. The pellets were then resuspended in a droplet of 1× PHEM buffer and taken up into cellulose capillary tubes. For each strain, the capillaries were cut into 3 mm pieces, placed in 6 mm type A specimen carrier (side 0.2 mm) filled with hexadecen, covered by 6 mm type B (side flat), and frozen with a high-pressure freezing machine (EM ICE, Leica Microsystems) as previously described (*Baquero et al., 2021*). The samples were then freeze-substituted in a mix of 1% $OsO4$, 5% $H_2O$ in pure acetone for 24 hr at –90°C, 12 hr at 5°C/hr at –30°C, 12 hr at –30°C, 3 hr at 10°C/h until 0°C and 1 hr at 0°C. All samples were washed in pure acetone and gradually infiltrated under agitation in low-viscosity Spurr resin/acetone mix from 30% to 100%, as previously described (*Hall et al., 2013*). Samples were then embedded in pure Spurr resin (EMS), followed by polymerization for 48 hr at 60°C. Ultrathin sections (70 nm) were cut with a Leica Ultracut S microtome and stained with 4% uranyl acetate and then 3% lead citrate. Transmission electron microscopy (TEM)

images were captured with a Tecnai Spirit 120 kV TEM equipped with a bottom-mounted Eagle 4k × 4k camera (FEI, USA). For statistical analysis, internal and external cell wall thickness have been measured (between 37 and 40 measurements, on 7–13 cells per condition) using TEM Imaging and Analysis software (TIA, FEI).

## Immunolabeling

Rabbit polyclonal antibodies were produced by ProteoGenix (Schiltigheim, France) using pustulan-conjugated BSA as the antigen. After 7–8 weeks of immunization, the resulting serum was used without antibodies purification. The specificity of serum was tested on pustulan, curdlan, and cell wall AI fractions from *Aspergillus fumigatus* (which did not contain β-1,6-glucan) and *C. albicans* by ELISA (not shown). For immunolabeling of β-1,6-glucans and β-1,3-glucans in the cell wall of *C. albicans*, cells were grown as described above. Then, 1 mL of culture was fixed at ambient temperature in the dark (1–2 hr) with paraformaldehyde 3.8% (PFA, Electron Microscopy Sciences). Cells were then centrifuged 3 min at 8000 × *g* and washed three times with 500 µL PBS. In each step, cells were resuspended vigorously to separate the cells and disperse aggregates. Then, cells were resuspended vigorously in 500 µL of PBS and 5% goat serum (Sigma) and then 20 µL were added to each well of a slide (Diagnostic Microscope Slides, 8-well 6 mm, Thermo Scientific) coated poly-L-lysine 0.1% and incubated 1 hr at ambient temperature to permit cell adhesion. Liquid was removed and PBS 1× and 5% goat serum was added again and incubated during 1 hr at ambient temperature. The liquid was aspirated and 20 µL of first antibody was added in PBS 1× and 5% goat serum overnight at 4°C (dilution 1/100 for polyclonal anti-β-1,6-glucan produced in rabbit, 1/250 for monoclonal anti-β-1,3-glucan [named 5H5] produced in mouse and kindly provided by N. Nifantiev) (*Matveev et al., 2019*). Wells were washed three times with PBS 1× and 5% goat serum. The second antibody was then added in PBS 1× and 5% goat serum 1 hr at ambient temperature (dilution 1/200), Alexa Fluor 488 goat anti-mouse IgG [AF488, Molecular Probes] or fluorescein goat anti-rabbit IgG [FITC, Invitrogen]. Wells were washed three times with PBS and 5% (v/v) goat serum and once with PBS. Then, one drop of Fluoromount-G (Invitrogen) was added to each well, wells were covered with a coverslip No. 1.5 (Menzel-Gläser), and polymerization was done during 48 hr. Cells were observed on an EVOS FL microscope (Life Technologies), with a magnification ×100 objective with oil immersion.

## PBMC and neutrophil isolation, stimulation by parietal fractions, and cytokine quantification

Human PBMCs from healthy donors were isolated by a density-gradient separation of Ficoll 400 (Eurobio, France). Isolated PBMCs were re-suspended in RPMI 1640 + GlutaMAX medium (Gibco), supplemented by 20% normal human sera (NHS, Normal Human Serum-Pooled, Zenbio) and seeded in each well (100 µL containing $4 \times 10^6$ cells/mL) of 96-well microtiter plates (TPP). Cell wall fractions suspended in RPMI 1640 + GlutaMAX medium were then added at a final concentration of 25 µg/mL. LPS (Sigma, 0.1 µg/mL) was used as a positive control. After 24 hr incubation at 37°C in a 5% $CO_2$ chamber (Cytoperm 2, Thermo Fisher Scientific), the culture supernatants were collected and stored at –20°C until further analysis.

Neutrophils from healthy donors were isolated with EasySep Direct Human Neutrophil Isolation Kit (STEMCELL Technologies) according to the manufacturer's instructions. Isolated neutrophils were re-suspended in RPMI 1640 + GlutaMAX medium (Gibco) and seeded in each well (100 µL containing $1 \times 10^6$ cells/mL) of 24-well microtiter plates (TPP). 200 µL of NHS at 3% diluted in RPMI 1640 + GlutaMAX medium were then added. Cell wall fractions suspended in RPMI 1640 + GlutaMAX medium were then added at a final concentration of 25 µg/mL. LPS (0.1 µg/mL) was used as a positive control. After 16 hr incubation at 37°C in a 5% $CO_2$ chamber (Cytoperm 2, Thermo Fisher Scientific), the culture supernatants were collected and stored at –20°C until further analysis.

Global cytokines, chemokines, and acute-phase proteins (total 36) present in the supernatants were detected with using Proteome Profiler Human Cytokines Array Kit (R&D Systems) according to the manufacturer's instructions. Then, the cytokines, chemokines, and acute-phase proteins of interest identified through protein profiler were quantified using DuoSet ELISA kits (R&D Systems) according to the manufacturer's instructions.

Native AI and periodate oxidized AI (AI-OxP) fractions and purified β-1,6-glucans from *C. albicans* were used. AI-OxP fraction was generated by the treatment of 40 mg of AI fraction with 100 mM

$m$-IO$_4$Na (4 mL), 3 days at 4°C. The reaction was stopped by the addition of 200 µL of glycerol, oxidized fraction was washed by centrifugation with water, reduced with BH$_4$Na (10 mg/mL in 0.1 mM NaOH, overnight) then submitted to mild acid hydrolysis (AcOH 10% 100°C, 1 hr), and finally washed with water and freeze-dried. Purified β-1,6-glucans were produced by digesting the AI fraction (40 mg) with LamA as described above. LamA was removed by passing the sample through a C-18 SPE cartridge (Sep-Pak classic, Waters) eluted with 5% acetonitrile and 0.1% TFA. Then, samples were purified by gel filtration chromatography (Superdex 200, 60 cm × 16 cm, Cytiva) in ammonium acetate 150 mM, pH 4 at a flow rate of 0.5 mL/min. Samples were detected by a refractive index detector (Iota2, Precision Instruments) and dialyzed against water and then freeze-dried. The fractions were quantified by the phenol-sulfuric acid colorimetric assay as previously described.

## Complement activation

Complement activation was done as previously described (**Wong et al., 2020**). Briefly, cell wall fraction from *C. albicans* – AI, AI-OxP, and purified β-1,6-glucans – were coated on 96-well microtiter plates (100 µL/well) at three different concentrations of 50 µg, 25 µg, and 12.5 µg per well in 50 mM bicarbonate buffer, pH 9.6, overnight at ambient temperature. Then, supernatants were discarded and wells were blocked with PBS 1×/BSA 1% for 1 hr at ambient temperature. Then, 8 µL of NHS and 92 µL of Gelatin-Veronal Buffer (GVB, 5 mM barbital, 145 mM NaCl, 0.1% gelatin, pH 7.4) supplemented by MgCl$_2$ (0.5 mM) and CaCl$_2$ (0.15 mM) were added, incubated for 1 hr, and then washed three times with PBS 1×/Tween-20 0.05%. Complement activation was measured by quantifying deposited C3b upon adding the wells with mouse monoclonal anti-human C3b antibody (MA1-82814, Thermo Fisher Scientific, diluted 1:1000) diluted in PBS 1×/BSA 1% and incubating 1 hr at ambient temperature, washing the wells with PBS 1×/Tween 20 0.05%, adding the wells with peroxidase-conjugated secondary anti-mouse IgG antibodies (Sigma-Aldrich, diluted 1:1000), and incubating for 1 hr at ambient temperature. After washing, 100 µL of substrate solution (TMB, BioFX; 100 µL/well) was added and the reaction was stopped with 50 µL of 4% H$_2$SO$_4$. The absorbance was read at 450 nm after subtraction of reference wavelength at 540 nm using microplate reader (Infinite m200 pro, TECAN).

## Generation of mutants by transient CRISPR/Cas9

Homozygous null mutants (*kre6Δ/Δ; skn1Δ/Δ; kre62Δ/Δ; skn2Δ/Δ; kre6/skn1Δ/Δ; kre6/skn1/kre62/skn2Δ/Δ*) were generated in *C. albicans* SC5314 using established transient CRISPR-Cas9 methods (**Vyas et al., 2015**; **Min et al., 2016**). An sgRNA cassette was constructed by PCR, first by synthesizing the SNR52 promoter (using primers SNR52/F and SNR52/R/GOI) and the sgRNA scaffold (with the primers sgRNA/F/GOI and sgRNA/R), using plasmid pV1093 as a template (**Vyas et al., 2015**). These two products were fused by PCR, followed by nested PCR with primers SNR52/N and sgRNA/N. The *CaCAS9* cassette was amplified from the plasmid pV1093 with primers CaCas9/F and CaCas9/R. Repair templates (RT), which contained the *SAT1*-Flipper marker and harbored 80 bp homology to the 5′ and 3′ ends of the target gene, were amplified from pSFS2A (**Reuss et al., 2004**). The RT containing the hygromycin resistance gene *HygB* was amplified from pCrispr-gRNA1/hygR, derived from plasmid pV1090 (**Vyas et al., 2015**) as follows: a *BspHI* fragment carrying the *HygB* marker was cut from pYM70 (**Basso et al., 2010**), the ends filled in with the Klenow enzyme and the fragment ligated into pV1090 digested with *StuI* and *BglII* and treated with the Klenow enzyme. PCR reactions to amplify sgRNA and CaCas9 were carried out using Phusion High-Fidelity DNA polymerase (NEB), and Q5 polymerase (NEB) was used for RT amplification, according to the manufacturer's instructions. All PCR products were cleaned up and concentrated with NucleoSpin Gel and PCR Clean-up (Macherey-Nagel). PCR products (3 µg of CaCAS9 cassette, 1 µg of sgRNA cassette, and 3 µg of the relevant RT) were transformed into *C. albicans* SC5314 using the lithium acetate method (**Sanglard et al., 1996**). Transformants were selected on YPD containing 150 µg/mL nourseothricin (Jena Bioscience) or YPD with 600 µg/mL hygromycin (Sigma). Disruption of both alleles of the target locus and integration of the RT were confirmed by PCR after DNA extraction (MasterPure Yeast DNA Purification Kit, Biosearch Technologies). The *SAT1* gene was removed with the flipper system. Briefly, the transformants were cultivated overnight in 3 mL YP with 2% maltose at 30°C. Then, the culture was diluted to 250 cells/mL in sterile water and 200 µL were plated on YPD with 25 µg/mL nourseothricin and incubated 1–2 days at 30°C. The smaller colonies were replicated on YPD vs. YPD + 150 µg/mL nourseothricin.

Transformants that grew only YPD were checked by PCR using primers Flanking_F/GOI and Flanking_R/GOI. The primers used for the construction and PCR check of these mutants are described in *Supplementary file 2*. Deletions were validated by PCR (*Figure 4—figure supplement 1*).

The quadruple mutant *kre6/skn1/kre62/skn2Δ/Δ* was complemented by integrating a StuI-linearized plasmid bearing the *KRE6* gene under the control of the constitutive promoter *ACT1* at the *RPS1* locus. This plasmid was obtained using the Gateway technology (Invitrogen) as described previously (*Chauvel et al., 2023*). *KRE6* was PCR amplified on SC5314 genomic DNA with primers KRE6-FWD and KRE6-REV; the fragment was cloned in pDONR207 (*Chauvel et al., 2012*) with the Invitrogen Gateway BP clonase, and then transferred to the destination vector CIp10-P$_{ACT1}$-SAT1 (*Chauvel et al., 2023*; *Legrand et al., 2018*) using the Invitrogen Gateway LR clonase. Successful cloning into the destination vector was confirmed by sequencing. Transformants were selected on YPD + 100 µg/mL nourseothricin. Plasmid integration at the *RPS1* locus was checked by colony PCR using CIpUL and CIpSAT.

## Phenotypic analyses

Kinetic curves were generated in 96-well plate (TPP) in SD medium, 30°C, at initial OD$_{620\,nm}$ = 0.01 using a Tecan Sunrise absorbance microplate reader. Optical density was measured every 10 min during 80 hr with Magellan Software. Doubling time was calculated from growth curves with GraphPad Prism software by generating a nonlinear regression (logistic curve) to determine the *k* coefficient, and then calculated as doubling time = ln(2)/k.

Spot test drug sensitivity assays were performed on complete SD medium agar plates containing either Congo Red (50 µg/mL), Calcofluor White (50 µg/mL), tunicamycin (1 µg/mL), nikkomycin (32 µg/mL), or caspofungin (0.5 µg/mL). Cells were precultured for 24 hr in SD medium liquid, 30°C, and then diluted to OD$_{600\,nm}$ = 1 in sterile water, and then tenfold serially diluted. Samples of 5 µL of each concentration were spotted onto agar plates. Pictures of plates were taken after 24 hr, 48 hr, 72 hr and 6 days with a Phenobooth+Colony counter (Singer Instruments). To observe macroscopic filamentation, 5 µL (OD$_{600\,nm}$ = 1) were spotted on YNB 2% GlcNAc, buffered at pH 7.2 with 100 mM MOPS agar, at 37°C, for 6 days. Pictures were taken using an iPhone camera.

To observe hyphal growth, cells were grown overnight in SD liquid 30°C, then diluted to OD$_{600\,nm}$ = 0.2 in 1 mL of 24-well plate (TPP) and grown for 6 hr either in SD, 30°C (control, no filamentation) or in YNB 2% GlcNAc, buffed at pH 7.2 with 100 mM MOPS, at 37°C (inducing filamentation). Cells were then observed with an Olympus IX 83 microscope, and images were captured with a Hamamatsu ORCA Flash cooled CCD camera using the cellSens imaging software.

## Statistical analyses

GraphPad Prism 10 software was used for statistical analyses. Data were generated from at least three independent biological replicates and then expressed as means ± standard deviation. Specific tests are indicated in the figure legends. Normally distributed datasets (QQ plot or Shapiro–Wilk test) were compared with ordinary one-way ANOVA (Dunnett's or Tukey's multiple comparisons test) or two-tailed unpaired *t*-test. For immunology data, measurements were paired/dependant as the same donors were exposed to different cell wall fractions, thus we applied the nonparametric test for non-normally distributed data: Friedman test (with Dunn's multiple comparisons test). p-values<0.05 were considered significant and represented on graphs as follows: *p<0.05; **p<0.01; ***p<0.001; ****p<0.0001; ns: nonsignificant.

## Acknowledgements

CB is the recipient of a PhD fellowship from the Laboratoire d'Excellence Integrative Biology of Emerging Infectious Diseases (ANR-10-LABX-62-IBEID). We acknowledge support from the French Government's Investissement d'Avenir program (Laboratoire d'Excellence Integrative Biology of Emerging Infectious Diseases [ANR-10-LABX-62-IBEID]) and Institut Pasteur from Paris. VA and TF are supported by the Agence Nationale de la Recherche ANR-21-CE17-0032-01, FUNPOLYVAC grant. We thank Romain Laurian and Natacha Sertour for their advice for molecular biology, Cécile Gautier for her advice on Prism software and spotting assays, Catherine Comte, Lucia Oreus, and Jamal Boutchlhit for material availability, and Matt Edmondson for providing *C. albicans* strains. The authors thank Suzan Noble, Mathias Richard, and Neil Gow for the access to their deletion mutant collections.

NG acknowledges support of Wellcome Trust Investigator, Collaborative, Equipment, Strategic and Biomedical Resource awards (101873, 200208, 215599, 224323). NG also thanks the MRC (MR/M026663/2) and the MRC Centre for Medical Mycology (MR/N006364/2) for support. This study/research was funded by the National Institute for Health and Care Research (NIHR) Exeter Biomedical Research Centre (BRC). The views expressed are those of the author(s) and not necessarily those of the NIHR or the Department of Health and Social Care. The 800-MHz NMR spectrometer of the Institut Pasteur was partially funded by the Région Ile de France (SESAME 2014 NMRCHR grant no. 4014526). CS is grateful for support for equipment from the French Government Programme Investissements d'Avenir France BioImaging (FBI, No. ANR-10-INSB-04-01).

## Additional information

### Funding

| Funder | Grant reference number | Author |
|---|---|---|
| Laboratoire d'Excellence Integrative Biology of Emerging Infectious Diseases | ANR-10-LABX-62-IBEID | Clara Bekirian<br>Sophie Bachellier-Bassi<br>Murielle Chauvel<br>Christophe d'Enfert<br>Thierry Fontaine |
| Agence Nationale de la Recherche | ANR-21-CE17-0032-01 | Vishu Kumar Aimanianda<br>Thierry Fontaine |
| Wellcome Trust | 10.35802/101873 | Neil AR Gow |
| Wellcome Trust | 10.35802/200208 | Neil AR Gow |
| Wellcome Trust | 10.35802/215599 | Neil AR Gow |
| Wellcome Trust | 10.35802/224323 | Neil AR Gow |
| MRC Centre for Medical Mycology | MR/N006364/2 | Neil AR Gow |
| Region Ile de France | SESAME 2014 NMRCHR grant no 4014526 | J Inaki Guijarro |
| French Government Programme Investissements d'Avenir France BioImaging | ANR-10-INSB-04-01 | Cyril Scandola |
| Medical Research Council | MR/M026663/2 | Neil AR Gow |
| Institut Pasteur | | Christophe d'Enfert<br>Thierry Fontaine<br>J Inaki Guijarro |
| National Institute for Health and Care Research | | Neil AR Gow |

The funders had no role in study design, data collection and interpretation, or the decision to submit the work for publication. For the purpose of Open Access, the authors have applied a CC BY public copyright license to any Author Accepted Manuscript version arising from this submission.

### Author contributions

Clara Bekirian, Conceptualization, Data curation, Formal analysis, Investigation, Methodology, Writing – original draft, Writing – review and editing; Isabel Valsecchi, Murielle Chauvel, Formal analysis, Methodology; Sophie Bachellier-Bassi, Conceptualization, Supervision, Methodology; Cyril Scandola, Thierry Mourer, Methodology; J Inaki Guijarro, Formal analysis, Visualization, Methodology, Writing – original draft; Neil AR Gow, Resources, Writing – original draft, Writing – review and editing; Vishu Kumar Aimanianda, Conceptualization, Resources, Formal analysis, Supervision, Validation, Methodology, Writing – original draft, Writing – review and editing; Christophe d'Enfert, Resources, Validation, Writing – original draft, Project administration, Writing – review and editing; Thierry Fontaine,

Conceptualization, Resources, Data curation, Formal analysis, Supervision, Funding acquisition, Validation, Investigation, Visualization, Methodology, Writing – original draft, Project administration, Writing – review and editing

### Author ORCIDs
Clara Bekirian ![ORCID] https://orcid.org/0009-0004-9300-949X
Cyril Scandola ![ORCID] https://orcid.org/0000-0002-5305-9095
Vishu Kumar Aimanianda ![ORCID] https://orcid.org/0000-0001-5813-7497
Thierry Fontaine ![ORCID] https://orcid.org/0000-0002-8184-789X

### Ethics
Human blood samples were taken from healthy donors from Etablissement Français du Sang Trinité (Paris, France) with written and informed consent, as per the guidelines provided by the institutional ethics committee, Institut Pasteur (convention 12/EFS/023).

Reviewer #1 (Public review): https://doi.org/10.7554/eLife.100569.3.sa1
Reviewer #2 (Public review): https://doi.org/10.7554/eLife.100569.3.sa2
Reviewer #3 (Public review): https://doi.org/10.7554/eLife.100569.3.sa3
Author response https://doi.org/10.7554/eLife.100569.3.sa4

## Additional files

### Supplementary files
• Supplementary file 1. *C. albicans* strains used in this study.
• Supplementary file 2. Primers used in this study.
• MDAR checklist

### Data availability
All data supporting the findings of this study are available as source data files linked to the corresponding figure or figure supplement.

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

# Appendix 1

**Appendix 1—key resources table**

| Reagent type (species) or resource | Designation | Source or reference | Identifiers | Additional information |
|---|---|---|---|---|
| Other | Yeast Nitrogen Base with amino acids (YNB) | BD | 239210 | Medium used in this study. See section 'Strains and growth conditions'. |
| Chemical compound, drug | Lactate | ChemCruz | sc-220120B | |
| Chemical compound, drug | $H_2O_2$ | Prolabo | 23612.294 | |
| Chemical compound, drug | MES | Sigma-Aldrich | M3058-100G | |
| Chemical compound, drug | MOPS | Sigma-Aldrich | M1254-1KG | |
| Chemical compound, drug | Acetate | Sigma-Aldrich | S2889-1KG | |
| Chemical compound, drug | Propionate | Sigma | P1880-100G | |
| Chemical compound, drug | Butyrate | Sigma-Aldrich | B5887-250MG | |
| Chemical compound, drug | Calcofluor white (CFW) | Sigma-Aldrich | 910090-20mL | |
| Chemical compound, drug | Congo Red (CR) | Sigma | C6277-25G | |
| Chemical compound, drug | Tunicamycin | Sigma | 17765-5MG | From *Streptomyces* spp. |
| Chemical compound, drug | Nikkomycin Z | Sigma | N-8028 | From *Streptomyces tendae* |
| Chemical compound, drug | Caspofungin | Sigma | SML0425-5MG | |
| Other | Yeast Nitrogen Base without amino acids | BD | 291940 | Medium used in this study for biofilm production. See section 'Strains and growth conditions'. |
| Other | Breathseal sealing membranes | Greiner Bio-one | 676051 | Used in this study for biofilm production. See section 'Strains and growth conditions'. |
| Commercial assay or kit | Pierce BCA Protein Assay Kit | Thermo Scientific | 23225 | |
| Chemical compound, drug | Trifluoroacetic acid (TFA) | Sigma | 8.08260.0101 | |
| Chemical compound, drug | Hydrochloric acid (HCl) | Fisher | H/1200/PC15 | |
| Chemical compound, drug | 4-Hydroxybenzhydrazide (PAHBAH) | Sigma | H9882-100G | |
| Chemical compound, drug | Sodium (meta)periodate (m-$IO_4Na$) | Sigma-Aldrich | S1878-25G | |
| Chemical compound, drug | Sodium sulfite | Sigma-Aldrich | S0505-250G | |
| Biological sample | Pustulan | Elicityl OligoTech | GLU900-1g | From *Lasallia pustulata* |
| Peptide, recombinant protein | LamA | PMID:9168619 | | From *Thermotoga neapolitana* expressed in *Escherichia coli* |

*Appendix 1 Continued on next page*

*Appendix 1 Continued*

| Reagent type (species) or resource | Designation | Source or reference | Identifiers | Additional information |
|---|---|---|---|---|
| Peptide, recombinant protein | Endo-β-1,6-glucanase | PMID:20852022 | | From *Schizosaccharomyces pombe* expressed in *Pichia pastoris* |
| Chemical compound, drug | Sodium azide (NaN₃) | Sigma-Aldrich | S-8032 | |
| Chemical compound, drug | $D_2O$ | Eurisotop | 17247446 | |
| Other | CarboPAC-PA1 column | Thermo Scientific | 035391 | See sections 'Cell wall polymers quantification', 'Quantification of β-1,6-glucan in AI and AS fractions,' and 'Structural characterization of β-1,6-glucan'. |
| Biological sample | Laminarin | Sigma | L9634-5G | From *Laminaria digitata* |
| Other | Superdex 200 column | Cytiva | 28990944 | See sections 'Structural characterization of β-1,6-glucan' and 'PBMC and neutrophil isolation, stimulation by parietal fractions and cytokine quantification'. |
| Biological sample | Dextran 6 kDa | Sigma-Aldrich | 31388-25G | From *Leuconostoc* spp. |
| Chemical compound, drug | Dextran 10, 40, 70, and 500 kDa | Pharmacia Fine Chemicals | | |
| Chemical compound, drug | Glutaraldehyde | Sigma-Aldrich | G6257-100ML | |
| Other | 6 mm type A and B specimen carriers | Leica | 16770181 and 16770182 | See section 'Transmission electron microscopy'. |
| Chemical compound, drug | Spurr resin | EMS | 15000 | |
| Antibody | Polyclonal anti-β-1,6-glucan (rabbit polyclonal) | ProteoGenix | This study | Pustulan-conjugated BSA used as the antigen IF (1:100) |
| Chemical compound, drug | Curdlan | Wako | 034-09901 | |
| Chemical compound, drug | Paraformaldehyde | Electron Microscopy Sciences | 15714 | |
| Biological sample | Goat serum | Sigma | G9023 | |
| Other | Diagnostic Microscope Slides, 8-well 6 mm | Thermo Scientific | ER-301B-CE24 | See section 'Immunolabeling'. |
| Chemical compound, drug | Poly-L-lysine | Sigma-Aldrich | P4707 | |
| Antibody | Monoclonal anti-β-1,3-glucan, named 5 H5 (mouse monoclonal) | PMID:31022215 | | IF (1:250) |
| Antibody | AlexaFluor 488 goat anti-mouse IgG (goat polyclonal) | Molecular Probes | A11029 | IF (1:200) |
| Antibody | Fluorescein goat anti-rabbit IgG (goat polyclonal) | Invitrogen | F2765 | IF (1:200) |
| Chemical compound, drug | Fluoromount-G | Invitrogen | 00-4958-02 | |
| Chemical compound, drug | Ficoll 400 | Eurobio | CMSMSL01-0U | |
| Chemical compound, drug | RPMI + GlutaMAX medium | Gibco | 61870044 | |

*Appendix 1 Continued on next page*

*Appendix 1 Continued*

| Reagent type (species) or resource | Designation | Source or reference | Identifiers | Additional information |
|---|---|---|---|---|
| Biological sample | NHS | Zenbio | HSER-ABP100ML | From human pooled donor |
| Biological sample | LPS | Sigma | L2630 | From *Escherichia coli* |
| Commercial assay or kit | EasySep Direct Human Neutrophil Isolation Kit | STEMCELL Technologies | 18001 | |
| Commercial assay or kit | Proteome Profiler Human Cytokines Array Kit | R&D Systems | ARY005B | |
| Commercial assay or kit | DuoSet ELISA kits | R&D Systems | DY201, DY278, DY206, DY217B, DY208, DY2037, DY271, DY279, DY210 | |
| Chemical compound, drug | BH$_4$Na | Aldrich | 213462-100G | |
| Chemical compound, drug | Glycerol | Riedel-deHaën | 15523 | |
| Chemical compound, drug | AcOH | Honeywell | 27221-1L | |
| Other | Sep-Pak classic C18 cartridge | Waters | WAT051910 | See section 'PBMC and neutrophil isolation, stimulation by parietal fractions and cytokine quantification'. |
| Peptide, recombinant protein | BSA | Sigma-Aldrich | A3059-100G | |
| Antibody | Anti-human C3b (mouse monoclonal) | Thermo Fisher | MA1-82814 | ELISA (1:1000) |
| Antibody | Peroxidase-conjugated secondary anti-mouse IgG (goat polyclonal) | Sigma-Aldrich | A4416 | ELISA (1:1000) |
| Chemical compound, drug | TMB | BioFX | TMBW-1000-01 | |
| Recombinant DNA reagent | pV1093 | PMID:25977940 | | |
| Recombinant DNA reagent | pSFS2A | PMID:5474295 | | |
| Recombinant DNA reagent | pV1090 | PMID:25977940 | | |
| Recombinant DNA reagent | pYM70 | PMID:20737428 | | |
| Peptide, recombinant protein | Phusion High-Fidelity DNA polymerase | NEB | M0530S | |
| Peptide, recombinant protein | Q5 polymerase | NEB | M0491S | |
| Commercial assay or kit | NucleoSpin Gel and PCR Clean-up | Macherey-Nagel | 740609.250 | |
| Chemical compound, drug | Nourseothricin | Jena Biosciences | AB-102 | |

*Appendix 1 Continued on next page*

*Appendix 1 Continued*

| Reagent type (species) or resource | Designation | Source or reference | Identifiers | Additional information |
|---|---|---|---|---|
| Commercial assay or kit | MasterPure Yeast DNA Purification Kit | Biosearch Technologies | MPY80200 | |
| Peptide, recombinant protein | BP clonase | Invitrogen | P/N 56480 | |
| Peptide, recombinant protein | LR clonase | Invitrogen | P/N 56485 | |
| Sequence-based reagent | pDONR207 | PMID:23049891 | | |
| Sequence-based reagent | Cip10-PACT1-SAT1 | PMID:29982705 | | |
| Strain | SC5314 | PMID:6394964 | | Clinical blood isolate |
| Strain | CAF2-1 | PMID:8349105 | | *URA3/ura3Δ::imm434* |
| Strain | CAI4 | PMID:8349105 | | *ura3Δ::imm434/ura3Δ::imm434* |
| Strain | BWP17 | PMID:10074081 | | *ura3Δ::imm434/ura3Δ::imm434 his1::hisG/his1::hisG arg4::hisG/arg4::hisG* |
| Strain | SN152 | PMID:15701792 | | *arg4Δ/arg4Δ leu2Δ/leu2Δ his1Δ/his1Δ URA3/ ura3Δ::imm434 IRO1/iro1Δ::imm434* |
| Strain | *cwh41Δ/Δ* | PMID:17933909 | | Same as CAI4 but *cwh41Δ::dp1200/ cwh41Δ::dp1200 RPS1/rps1Δ::Clp10* |
| Strain | *rot2Δ/Δ* | PMID:17933909 | | Same as CAI4 but *rot2Δ::dp1200/rot2Δ::dp1200 RPS1/rps1Δ::Clp10* |
| *Strain* | *kre5Δ/Δ* | PMID:20543849 | | *Same as SN152 but kre5Δ::leu2/kre5Δ::his1* |
| *Strain* | *kre6Δ/Δ* | This study | | *kre6Δ::HygB/kre6Δ::HygB* |
| Strain | *kre62Δ/Δ* | This study | | *kre62Δ::FRT/kre62Δ::FRT* |
| Strain | *skn2Δ/Δ* | This study | | skn2Δ::FRT/skn2Δ::FRT |
| Strain | *skn1Δ/Δ* | This study | | *skn1Δ::FRT/skn1Δ::FRT* |
| Strain | *kre6/skn1Δ/Δ* | This study | | *kre6Δ::HygB/kre6Δ::HygB skn1Δ::FRT/skn1Δ::FRT* |
| Strain | *kre6/kre62/skn2/skn1Δ/Δ* | This study | | *kre6Δ::HygB/kre6Δ::HygB kre62Δ::FRT/kre62Δ::FRT skn2Δ::FRT/skn2Δ::FRT skn1Δ::FRT/skn1Δ::FRT* |
| Strain | *kre6/kre62/skn2/skn1Δ/Δ+P_{ACT1}-KRE6* | This study | | *kre6Δ::HygB/kre6Δ::HygB kre62Δ::FRT/kre62Δ::FRT skn2Δ::FRT/skn2Δ::FRT skn1Δ::FRT/skn1Δ::FRT RPS1/RPS1::ClpSAT1-P_{ACT1}-KRE6* |
| Strain | *kre1Δ/Δ* | Provided by Mathias Richard | | Same as BWP17 but *kre1Δ::arg4/kre1Δ::his1* |
| *Strain* | *P_{MRP1}-CHS1/chs1Δ* | PMID:11251855 | | Same as CAI4 but *chs1Δ::hisG/chs1Δ:pSK-URA3-P_{MRP1}-CHS1* |
| *Strain* | *chs2Δ/Δ* | PMID:8636047 | | Same as CAF2-1 but *chs2Δ::hisG/chs2Δ::hisG-URA3-hisG* |
| Strain | *chs3Δ/Δ* | PMID:7479842 | | Same as CAF2-1 but *chs3-2::hisG/chs3-3::hisG-URA3-hisG* |
| Strain | *mnt1/mnt2Δ/Δ* | PMID:15519997 | | Same as CAF2-1 but mnt1-*mnt2Δ::hisG/mnt1-mnt2Δ::hisG-URA3-hisG* |

*Appendix 1 Continued on next page*

*Appendix 1 Continued*

| Reagent type (species) or resource | Designation | Source or reference | Identifiers | Additional information |
|---|---|---|---|---|
| Strain | *mnn2/22/21/23/24/26Δ/Δ* | PMID:23633946 | | Same as CAF2-1 but *mnn2Δ::dpl200/ mnn2Δ::dpl200 mnn22Δ::dpl200/mnn2Δ::dpl200 mnn23Δ::dpl200/mnn23Δ::dpl200 mnn24Δ::dpl200/ mnn24Δ::dpl200 mnn26Δ::dpl200/mnn26Δ::dpl200 mnn21Δ::dpl200/mnn21Δ::dpl200* |
| Strain | *mnn9Δ/Δ* | PMID:10601199 | | Same as CAF2-1 *mnn9Δ::hisG/mnn9Δ::hisG Δura3Δ::imm434/ura3Δ::imm434* |
| Strain | *och1Δ/Δ* | PMID:20543849 | | Same as SN152 but *och1Δ::leu2/och1Δ::his1* |
| Strain | *fks1Δ* | PMID:30370375 | | SC5314, but *fks1/fks1Δ* |
| Strain | *phr1Δ/Δ* | PMID:7823929 | | Same as BWP17 but *phr1Δ::hisG/phr1Δ* |
| Strain | *phr2Δ/Δ* | PMID:9315654 | | Same as BWP17 but *phr2Δ::hisG/phr2Δ::hisG-URA3-hisG* |
| Sequence-based reagent | SNR52/F | PMID:27340698 | PCR primers | See sequence in *Supplementary file 2* |
| Sequence-based reagent | sgRNA/R | PMID:27340698 | PCR primers | See sequence in *Supplementary file 2* |
| Sequence-based reagent | SNR52/N | PMID:27340698 | PCR primers | See sequence in *Supplementary file 2* |
| Sequence-based reagent | sgRNA/N | PMID:27340698 | PCR primers | See sequence in *Supplementary file 2* |
| Sequence-based reagent | CaCas9/F | PMID:27340698 | PCR primers | See sequence in *Supplementary file 2* |
| Sequence-based reagent | CaCas9/R | PMID:27340698 | PCR primers | See sequence in *Supplementary file 2* |
| Sequence-based reagent | SNR52/R/SKN1 | This study | PCR primers | See sequence in *Supplementary file 2* |
| Sequence-based reagent | SNR52/R/SKN2 | This study | PCR primers | See sequence in *Supplementary file 2* |
| Sequence-based reagent | SNR52/R/KRE62 | This study | PCR primers | See sequence in *Supplementary file 2* |
| Sequence-based reagent | SNR52/R/KRE6 | This study | PCR primers | See sequence in *Supplementary file 2* |
| Sequence-based reagent | sgRNA/F/SKN1 | This study | PCR primers | See sequence in *Supplementary file 2* |
| Sequence-based reagent | sgRNA/F/SKN2 | This study | PCR primers | See sequence in *Supplementary file 2* |
| Sequence-based reagent | sgRNA/F/KRE62 | This study | PCR primers | See sequence in *Supplementary file 2* |
| Sequence-based reagent | sgRNA/F/KRE6 | This study | PCR primers | See sequence in *Supplementary file 2* |
| Sequence-based reagent | SAT1FLP/F/SKN1 | This study | PCR primers | See sequence in *Supplementary file 2* |
| Sequence-based reagent | SAT1FLP/F/SKN2 | This study | PCR primers | See sequence in *Supplementary file 2* |
| Sequence-based reagent | SAT1FLP/F/KRE62 | This study | PCR primers | See sequence in *Supplementary file 2* |
| Sequence-based reagent | HygR/F/KRE6 | This study | PCR primers | See sequence in *Supplementary file 2* |

*Appendix 1 Continued on next page*

*Appendix 1 Continued*

| Reagent type (species) or resource | Designation | Source or reference | Identifiers | Additional information |
|---|---|---|---|---|
| Sequence-based reagent | SAT1FLP/R/SKN1 | This study | PCR primers | See sequence in *Supplementary file 2* |
| Sequence-based reagent | SAT1FLP/R/SKN2 | This study | PCR primers | See sequence in *Supplementary file 2* |
| Sequence-based reagent | SAT1FLP/R/KRE62 | This study | PCR primers | See sequence in *Supplementary file 2* |
| Sequence-based reagent | hygR/R/KRE6 | This study | PCR primers | See sequence in *Supplementary file 2* |
| Sequence-based reagent | Flanking_F/SKN1 | This study | PCR primers | See sequence in *Supplementary file 2* |
| Sequence-based reagent | Flanking_F/SKN2 | This study | PCR primers | See sequence in *Supplementary file 2* |
| Sequence-based reagent | Flanking_F/KRE62 | This study | PCR primers | See sequence in *Supplementary file 2* |
| Sequence-based reagent | Flanking_R/SKN1 | This study | PCR primers | See sequence in *Supplementary file 2* |
| Sequence-based reagent | Flanking_R/SKN2 | This study | PCR primers | See sequence in *Supplementary file 2* |
| Sequence-based reagent | Flanking_R/KRE62 | This study | PCR primers | See sequence in *Supplementary file 2* |
| Sequence-based reagent | screenKre6/F | This study | PCR primers | See sequence in *Supplementary file 2* |
| Sequence-based reagent | screenKre6/R | This study | PCR primers | See sequence in *Supplementary file 2* |
| Sequence-based reagent | ClpUL | This study | PCR primers | See sequence in *Supplementary file 2* |
| Sequence-based reagent | ClpSAT | This study | PCR primers | See sequence in *Supplementary file 2* |
| Sequence-based reagent | KRE6-FWD | This study | PCR primers | See sequence in *Supplementary file 2* |
| Sequence-based reagent | KRE6-REV | This study | PCR primers | See sequence in *Supplementary file 2* |

