## [Editor Report · eLife Assessment]

The article will be of broad interest to fungal biologists and fungal immunologists seeking to understand the biosynthesis of the fungal cell wall, in particular of ß-1,6-glucan synthesis and the importance of this so far understudied constituent of the cell wall for cell wall integrity and immune response. The study is of **fundamental** significance and adds structural clarity to the genetic and biochemical basis of this difficult-to-analyze carbohydrate. It opens the potential for understanding its role in immune recognition and potentially as a drug target. Overall, the data is **compelling**, properly controlled, and analyzed.

---

## [Referee Report · Reviewer #1 (Public review)]

Summary:

The fungal cell wall is a very important structure for the physiology of a fungus but also for the interaction of pathogenic fungi with the host. Although a lot of knowledge on the fungal cell wall has been gained, there is lack of understanding of the meaning of ß-1,6-glucan in the cell wall. In the current manuscript, the authors studied in particular this carbohydrate in the important human-pathogenic fungus Candida albicans. The authors provide a comprehensive characterization of cell wall constituents under different environmental and physiological conditions, in particular of ß-1,6-glucan. Also, β-1,6-glucan biosynthesis was found to be likely a compensatory reaction when mannan elongation was defective. The absence of β-1,6-glucan resulted in a significantly sick growth phenotype and complete cell wall reorganization. The manuscript contains a detailed analysis of the genetic and biochemical basis of ß-1,6-glucan biosynthesis which is apparently in many aspects similar to yeast. Finally, the authors provide some initial studies on immune modulatory effects of ß-1,6-glucan.

---

## [Referee Report · Reviewer #2 (Public review)]

Summary:

The authors provide the first (to my knowledge) detailed characterization of cell wall b-1,6 glucan in the pathogen Candida albicans. The approaches range from biochemistry to genetics to immunology. The study provides fundamental information and will be a resource of exceptional value to the field going forward. Highlights include the construction of a mutant that lacks all b-1,6 glucan and the characterization of its cell wall composition and structure. Figure 5a is a feast for the eyes, showing that b-1,6 glucan is vital for the outer fibrillar layer of the cell wall. Also much appreciated was the summary figure, Figure 7, that presents the main findings in digestible form.

Strengths:

The work is highly significant for the fungal pathogen field especially, and more broadly for anyone studying fungi, antifungal drugs, or antifungal immune responses.

The manuscript is very readable, which is important because most readers will be cell wall nonspecialists.

The authors construct a key quadruple mutant, which is not trivial even with CRISPR methods, and validate it with a complemented strain. This aspect of the study sets the bar high.

The authors develop new and transferable methods for b-1,6 glucan analysis.

Weaknesses:

The one "famous" cell type that would have been interesting to include is the opaque cell. Please include it in the next paper!

---

## [Referee Report · Reviewer #3 (Public review)]

Summary:

The cell wall of human fungal pathogens, such as Candida albicans, is crucial for structural support and modulating the host immune response. Although extensively studied in yeasts and molds, the structural composition has largely focused on the structural glucan b,1,3-glucan and the surface exposed mannans, while the fibrillar component β-1,6-glucan, a significant component of the well wall, has been largely overlooked. This comprehensive biochemical and immunological study by a highly experienced cell wall group provides a strong case for the importance of β-1,6-glucan contributing critically to cell wall integrity, filamentous growth, and cell wall stability resulting from defects in mannan elongation. Additionally, β-1,6-glucan responds to environmental stimuli and stresses, playing a key role in wall remodeling and immune response modulation, making it a potential critical factor for host-pathogen interactions.

Strengths:

Overall, this study is well designed and executed. It provides the first comprehensive assessment of β-1,6-glucan as a dynamic, albeit underappreciated, molecule. The role of β-1,6-glucan genetics and biochemistry has been explored in molds like Aspergillus fumigatus, but this work shines important light on its role in Candida albicans. This is important work that is of value to Medical Mycology, since β-1,6-glucan plays more than just a structural role in the wall. It may serve as a PAMP and a potential modulator of host-pathogen interactions.

Weaknesses:

In keeping with an important role in immune recognition, it was suggested that the manuscript rigor would benefit from a more physiological evaluation ex vivo and preferably in vivo, assessment on stimulating the immune system within in the cell wall and not just as a purified component. This is a critical outcome measure for this study and gets squarely at its importance for host-pathogen interactions, especially in response to environmental stimuli and drug exposure. The authors addressed this issue contextually and indicate that it will require a more detailed immunologic evaluation but is not in keeping with the intent of this foundational study.

---

## [Author Response]

The following is the authors’ response to the original reviews.

**Public Reviews:**

**Reviewer #1 (Public Review):**
Summary:The fungal cell wall is a very important structure for the physiology of a fungus but also for the interaction of pathogenic fungi with the host. Although a lot of knowledge on the fungal cell wall has been gained, there is a lack of understanding of the meaning of ß-1,6-glucan in the cell wall. In the current manuscript, the authors studied in particular this carbohydrate in the important humanpathogenic fungus Candida albicans. The authors provide a comprehensive characterization of cell wall constituents under different environmental and physiological conditions, in particular of ß-1,6glucan. Also, β-1,6-glucan biosynthesis was found to be likely a compensatory reaction when mannan elongation was defective. The absence of β-1,6-glucan resulted in a significantly sick growth phenotype and complete cell wall reorganization. The manuscript contains a detailed analysis of the genetic and biochemical basis of ß-1,6-glucan biosynthesis which is apparently in many aspects similar to yeast. Finally, the authors provide some initial studies on the immune modulatory effects of ß-1,6-glucan.Strengths:The findings are very well documented, and the data are clear and obtained by sophisticated biochemical methods. It is impressive that the authors successfully optimized methods for the analyses and quantification of ß-1-6-glucan under different environmental conditions and in different mutant strains.Weaknesses:However, although already very interesting, at this stage there are some loose ends that need to be combined to strengthen the manuscript. For example, the immunological studies are rather preliminary and need at least some substantiation. Also, at this stage, the manuscript in some places remains a bit too descriptive and needs the elucidation of potential causalities.
**Reviewer #2 (Public Review):**
Summary:The authors provide the first (to my knowledge) detailed characterization of cell wall b-1,6 glucan in the pathogen Candida albicans. The approaches range from biochemistry to genetics to immunology. The study provides fundamental information and will be a resource of exceptional value to the field going forward. Highlights include the construction of a mutant that lacks all b-1,6 glucan and the characterization of its cell wall composition and structure. Figure 5a is a feast for the eyes, showing that b-1,6 glucan is vital for the outer fibrillar layer of the cell wall. Also much appreciated was the summary figure, Figure 7, which presents the main findings in digestible form.Strengths:The work is highly significant for the fungal pathogen field especially, and more broadly for anyone studying fungi, antifungal drugs, or antifungal immune responses.The manuscript is very readable, which is important because most readers will be cell wall nonspecialists.The authors construct a key quadruple mutant, which is not trivial even with CRISPR methods, and validate it with a complemented strain. This aspect of the study sets the bar high. The authors develop new and transferable methods for b-1,6 glucan analysis.Weaknesses:The one "famous" cell type that would have been interesting to include is the opaque cell. This could be included in a future paper.
**Reviewer #3 (Public Review):**
Summary:The cell wall of human fungal pathogens, such as Candida albicans, is crucial for structural support and modulating the host immune response. Although extensively studied in yeasts and molds, the structural composition has largely focused on the structural glucan b,1,3-glucan and the surface exposed mannans, while the fibrillar component β-1,6-glucan, a significant component of the well wall, has been largely overlooked. This comprehensive biochemical and immunological study by a highly experienced cell wall group provides a strong case for the importance of β-1,6-glucan contributing critically to cell wall integrity, filamentous growth, and cell wall stability resulting from defects in mannan elongation. Additionally, β-1,6-glucan responds to environmental stimuli and stresses, playing a key role in wall remodeling and immune response modulation, making it a potential critical factor for host-pathogen interactions.Strengths:Overall, this study is well-designed and executed. It provides the first comprehensive assessment of β-1,6-glucan as a dynamic, albeit underappreciated, molecule. The role of β-1,6-glucan genetics and biochemistry has been explored in molds like Aspergillus fumigatus, but this work shines an important light on its role in Candida albicans. This is important work that is of value to Medical Mycology, since β-1,6-glucan plays more than just a structural role in the wall. It may serve as a PAMP and a potential modulator of host-pathogen interactions. In keeping with this important role, the manuscript rigor would benefit from a more physiological evaluation ex vivo and preferably in vivo, assessment on stimulating the immune system within in the cell wall and not just as a purified component. This is a critical outcome measure for this study and gets squarely at its importance for host-pathogen interactions, especially in response to environmental stimuli and drug exposure.

**Response to reviewers (Public reviews):**

We thank all the three reviewers for their opinion on our work on *Candida albicans* β-1,6-glucan, which highlights the importance of this cell wall component in the biology of fungi. Here are our responses to their comments for public reviews:

(1) Indeed, the data presented for immunological studies is preliminary. It has been acknowledged by the reviewers that our analysis providing insights into the biosynthetic pathways involved in comprehensive in dealing with organization and dynamics of the β-1,6-glucan polymer in relation with other cell wall components and environmental conditions (temperature, stress, nutrient availability, etc.). However, we anticipated that there would be immediate curiosity as to what the immunological contribution of β-1,6 glucan and we therefore felt we needed to initiative these studies and include them. We therefore performed immunological studies to assess whether β-1,6-glucans act as a pathogen-associated molecular pattern (PAMP), and if so, what its immunostimulatory potential is. Our data clearly suggest that β-1,6-glucan is a PAMP, and consequently lead to several questions: (a) what are the host immune receptors involved in the recognition of this polysaccharide, and thereby the downstream signaling pathways, (b) how is β-1,6-glucan differentially recognized by the host when *C. albicans* switches from a commensal to an opportunistic pathogen, and (c) how does the host environment impact the exposure of this polysaccharide on the fungal surface. We believe addressing these questions is beyond the scope of the present manuscript and aim to present new data in future manuscript. Nonetheless, in the revised manuscript, suggest approaches that we can take to identify the receptor that could be involved in the recognition of β-1,6-glucan. Moreover, we have modified the discussion presenting it based on the data rather than being descriptive.

(2) It will be interesting to assess the organization of β-1,6-glucan and other cell wall components in the opaque cells. It is documented that the opaque cells are induced at acidic pH and in the presence of *N*-acetylglucosamine and CO2. Our data shows that pH has an impact on β-1,6-glucan, which suggests that there will be differential organization of this polysaccharide in the cell wall of opaque cells. As suggested by the reviewer, we will include analysis of opaque cells (and other *C. albicans* cell types) in future studies.

With the exception of these major new avenues for this research, our revision can address each of the comments provided by the reviewers.

**Recommendations for the authors**

**Reviewer #1 (Recommendations For The Authors):**
Although the study is very interesting, there are some loose ends that need to be combined to strengthen the manuscript. For example, the immunological studies are rather preliminary and need at least some substantiation. Also, at this stage, the manuscript in some places remains a bit too descriptive and needs the elucidation of potential causalities.Specifically:(1) As you showed, defects in chitin content led to a decrease in the cross-linking of β-glucans in the inner wall that corresponded to the effect of nikkomycin-treated C. albicans phenotype; conversely, an increase in chitin content led to more cross-linking of β-glucans as observed in the FKS1 mutant or in the presence of caspofungin. What is the mechanistic reason for these observations?

On one hand, yeast cell wall chitin occurs in three forms: free and covalently linked to β-1,3-glucan or β-1,6-glucan; crosslinked β-glucan-chitin forms core fibrillar structure resistant to alkali. A decrease in the chitin content, therefore, affect β-glucan-chitin crosslinking thereby making β-glucan alkali-soluble. On the other hand, a decrease in the β-glucan content, as in *FKS1* mutant or upon caspofungin treatment, results in increased cell wall chitin and β-glucan-chitin contents. A decrease in the β-1,3-glucan biosynthesis is associated with upregulation of *CRH1* involved in the β-glucan-chitin crosslinking, which explains an increased β-glucan-chitin content in the *FKS1* mutant or upon caspofungin treatment. We have included in this discussion in the revised manuscript (p14, lines 2-10).

(2) The β-1,6-glucan biosynthesis is stimulated via a compensatory pathway when there is a defect in O- and N-linked cell wall mannan biosynthesis. Why? causality? Hypothesis?

Two phenomena were observed related to β-1,6-glucan and mannan biosynthesis: (1) a defect in the elongation of N-mannan led to an increase in the β-1,6-glucan content; (2) a defect of O-mannan elongation resulted in the reduce size of β-1,6-glucan chains, however, increased their branching. These observations of our study suggest a global rescue program of the cell wall damage that could occur due to defect in one of the cell wall contents. We have discussed this in the revised manuscript (p14, last paragraph, p15 first paragraph). Moreover, β-1,3-glucan and chitin are synthesized by respective membrane bound synthases, and a defect in of their synthesis is compensated by the other. In line, although need to be validated for β-1,6-glucan, biosynthesis of mannan and β-1,6-glucan seem to initiate intracellularly. Therefore, possibility is that the defective mannan biosynthesis could be compensated by β-1,6-glucan biosynthesis, but need to be further validated experimentally.

(3) You showed that the removal of β-1,6-glucan by periodate oxidation (AI-OxP) led to a significant decrease in the IL-8, IL-6, IL-1β, TNF-α, C5a, and IL-10 released, suggesting that their stimulation was in part β-1,6-glucan dependent. What is the consequence of the stimulation, e.g. better phagocytosis, etc.? This needs some more experiments, otherwise the data is purely descriptive, as the conclusion. Also, what do you want to show with the activation of the complement system? Is ß1,6-glucan detected by complement receptors? I think this is really a loose end. I think it is necessary to provide more data on this observation, which I think lacks control with serum lacking complement, this should then be moved to the main manuscript.

In this study, our aim was to assess whether β-1,6-glucan acts as a pathogen-associated molecular pattern (PAMP) of *C. albicans*, and if yes, what is its immunostimulatory capacity/potential. Our data confirms that, indeed, β-1,6-glucan acts as a PAMP, and its removal significantly reduces the immunostimulatory capacity of the fibrillar core structure of the *C. albicans* cell wall. On the other hand, data provided in the revised manuscript (see updated Figure S14, discussion p13 lines 16-21) indicate that the human serum factors significantly enhance the immunostimulatory capacity of β1,6-glucan and that β-1,6-glucan interacts with the complement component C3b. However, addressing the role of β-1,6-glucan in phagocytosis using β-1,6-glucan deletion mutant will not be possible as the cell wall of this mutant is modified, and β-1,6-glucan is not the only cell wall component interacting with C3b. Alternate is to coat β-1,6-glucan on beads and use to study phagocytosis and identify immune receptors; however, these are beyond the scope of our present study/focus.

(4) Also, you suggested that β-1,6-glucan and β-1,3-glucan stimulate innate immune cells in distinct ways. Please provide more data on this interesting suggestion. You can block the dectin-1 receptor for example or use dectin-1 deficient macrophages from mice. The part on the immune stimulation needs to be optimized.

Stimulation of immune cells by pustulan (insoluble linear β-1,6-glucan) via a dectin-1independent pathway has been described previously (PMIDs: 18005717, 16371356) as discussed in the manuscript. Our preliminary data indicate that dectin-1 blocking on immune cells (using antidectin-1 antibodies) has no effect on the immunostimulatory potential of β-1,6-glucan, unlike AI and AI-OxP that showed significantly reduced cytokine secretion by the immune cells upon dectin-1 blocking. Deciphering the β-1,6-glucan recognition and its immunomodulatory pathways are underway, and will be the subject of our future study/manuscript.

(5) β-1,6-glucan and mannan productions are coupled. What is the hypothesis? Is it due to the necessity of mannan residues in ß-1,6-glucan biosynthesis enzymes from the ER? Can that be experimentally proven?

β-1,6-glucan and mannan synthesis should be coupled in two ways. First, as mentioned above (Response 2), defects in mannan elongation led to an alteration of β-1,6-glucan production. Second, early steps of N-glycosylation led to a strong reduction of β-1,6-glucan size and its cell wall content. However, we do not believe that the synthesis of N-glycan is required for the synthesis of an acceptor essential to β-1,6-glucan synthesis. Defect in N-mannan elongation led to a global cell wall remodeling as described above. Kre5, Rot2 and Cwh41 are part of the calnexin cycle involved in the control of N-glycoprotein folding in the ER, suggesting that some protein directly involved in the β-1,6-glucan synthesis required a folding quality control to be active. We modified our discussion, accordingly, highlighting these points (p14, last paragraph, p15 second paragraph).

(6) As PHR1 and PHR2 genes are strongly regulated by external pH, the compensatory differences described may be explained by pH-dependent regulation of β-1,6-glucan synthesis.' Please check. Also, could the pH regulation form the basis of e.g. differences you found for ß-1,6-glucan under different environmental conditions, i.e., growth on different carbon sources leads to different external pH values, as shown for many fungi?

We agree that environmental pH is dependent on carbon source and pH varies during growth curve. To test the effect of pH we buffered the medium with 100 mM MOPS or MES. Clearly, Fig. 2 and S1 show that the pH has an effect on the cell wall composition and polymer exposure as previously described (PMID:28542528). Here, we show that pH has an impact on the β-1,6-glucan size as well as its branching. However, in buffered medium, addition of organic acid (such as acetate, propionate, butyrate or lactate) had an impact on cell wall composition, showing that not only pH has an effect on cell wall composition. About *phr1*Δ/Δ and *phr2*Δ/Δ mutants, we believe that the difference in the cell wall composition observed between mutants is mainly due to the pH-dependent regulation, which we indicated in the discussion (p14, end of first paragraph).

Minor:(1) In Figure 7B: dynamism should be replaced by dynamic and in term is rather in terms.

Modified as suggested.

(2) Replace molecular size with molecular mass when you give daltons.

Molecular size has been replaced by molecular weight, when presented as daltons.

(3) Page 7: for explanation, please add that nikkomycin is a chitin biosynthesis inhibitor.

As suggested, explained that nikkomycin is a chitin biosynthesis inhibitor.

**Reviewer #2 (Recommendations For The Authors):**
(1) I wondered if the increased chitin content of hyphae might reflect growth on the precursor GlcNAc. Have you tested hyphae that are induced in other ways? (2) Related to point 1, did you look at the relative abundance of yeast vs hyphae in the preparation? I wonder if yeast contamination might have reduced the extent of the composition changes observed.

We used GlcNAc as hyphae inducer as: (1) in presence of GlcNAc, hyphae are produced without any yeast contamination; in this condition, we observed an increase in the chitin content, as described, in hyphae (PMID:16423067); (2) we excluded using of serum, another condition inducing hyphal formation, as we could not control serum factors that may impact cell wall composition. We now indicate in the methods section that hyphae induced by GlcNAc were not contaminated by yeast (p17, line 3).

(3) I recommend rephrasing the first sentence of the Figure 2 legend: "Cells were grown in liquid SD medium at 37oC at exponential phase under different growth conditions." The conditions varied extensively - stationary is not exponential; biofilm is probably not exponential. Also, the "D" in "SD" stands for dextrose, and the carbon source varied a good deal. Perhaps you could say: "Cells were grown in liquid synthetic medium at 37oC under different growth conditions, as specified in Methods."

Sentences have been rephrased.

(4) Figure 7b has a typo: "dependant" for "dependent".

Typo-error has been corrected.

**Reviewer #3 (Recommendations For The Authors):**
To explore the biochemical composition of the cell wall, the authors fractionated the wall component into three categories based on polymer properties and reticulations: sodium-dodecyl-sulphate-βmercaptoethanol (SDS-β-ME) extract, alkali-insoluble (AI), and alkali-soluble (AS) fractions, and they developed several independent methods to distinguish between β-1,3-glucans and β-1,6-glucans. The composition and surface exposure of fungal cell wall polymers is known to depend on environmental growth conditions. It was shown that the cell wall of C. albicans hyphae increased chitin content (10% vs. 3%) and decreased β-1,6-glucan (18% vs. 23%) and mannan (13% vs. 20%) compared to the yeast form, and the reduced β-1,6-glucan content was associated with a smaller β1,6-glucan size (43 vs. 58 kDa), suggesting that both the content and structure of β-1,6-glucan are regulated during growth and cellular morphogenesis. Similar behavior was observed when exposing cells to acid and neutral medium pH. The most significant cell wall alteration occurred in a lactatecontaining medium, which led to a sharp reduction in structural core polysaccharides: chitin (-43%), β-1,3-glucan (-48%), and β-1,6-glucan (-72%). This reduction aligns with the previously observed decreases in inner cell wall layer thickness. As expected, the authors found that modulating chitin content genetically (chs3Δ/Δ knockout mutant) led to an increase of both β-1,3-glucan and β-1,6glucan. An increase in chitin content following genetic alteration of FKS genes impacting glucan synthase or after exposure to the echinocandin caspofungin led to enhanced cross-linking of βglucans. A slight increase in the β-1,3-glucan branching was also observed in the mnt1/mnt2Δ/Δ double mutant, suggesting that β-1,6-glucan and mannan synthesis may be coupled.- This effect is not that pronounced, and the relationship appears somewhat overstated and may reflect an indirect interaction. The authors should address accordingly.

We agree that this sentence was overstated. To make it clearer and less pronounced, we divided this sentence into to two with less pronounced statements (p8, line 34).

The genetics of β-1,6-glucan biosynthesis appear complex and a figure describing putative roles for specific genes would be beneficial. For example, KRE6 is a glucosyl hydrolase required for beta1,6-glucan biosynthesis.- It would be valuable to better understand the overall biosynthetic process. Please elaborate more in a figure.

Although proteins/enzymatic activities directly involved in the β-1,6-glucan biosynthesis have not yet been identified, as suggested by this reviewer, we included a schematic representation of this process based on our hypothesis (Figure S15, and p15 lines 17-22 in revised manuscript), indicating the possible involvement of Kre6p.

The deletion of KRE6 homologs, essential for β-1,6-glucan biosynthesis, resulted in the absence of β-1,6-glucan production, and significant structural alterations of the cell wall. This result nicely confirms the important role of β-1,6-glucan in regulating cell wall homeostasis. The absence of β1,6-glucan was associated with increased (mutant v. WT) chitin content (9.5% vs. 2.5%) and highly branched β- β-1,6-glucan 1,3-glucan (48% vs. 20%). TEM ultrastructure studies nicely showed the change in cell wall overall architecture. From a drug discovery perspective, since the blockade of β1,6-glucan did not block growth, it may have more value as a potential virulence target. This would be valuable but needs to be assessed in animal model challenge competition experiments.- The authors may want to elaborate more.

We agree and modified “antifungal target” as “potential virulence target”.

It is well known that β-1,3-glucan, mannan, and chitin function serve as PAMPs, which induce immune responses. The role of β-1,6-glucan as a PAMP is not well understood, and the authors provide evidence that different cell wall extracted fractions with enriched constituents induce immune responses invoking cytokines, chemokines, and acute phase proteins, as well as the complement system. While this data clearly shows that β-1,6-glucan is immunologically active and potentially important for host-pathogen interactions, the analysis is preliminary and falls short of making this case.- This is a critical point in getting at the potential host signaling of β-1,6-glucan contained in the cell wall or shed by the cell (is this known?)- This analysis would be bolstered significantly by examining stimulation relative to other cell wall components, and most importantly, whole cell modulation of β-1,6-glucan exposure for immune presentation, and not just unnatural concentrated extracts. This can be readily accomplished with the various mutants in hand, as well as after exposure to various antifungal agents echinocandins and nikkomycins (see Hohl et al. 2008 JID). Additional validation would benefit from animal model studies to examine in vivo immune modulation.

We agree with the reviewer. However, the main focus of our present work was to study the organization and dynamics of *C. albicans* cell wall β-1,6-glucan, and to explore its possible role as pathogen-associated molecular pattern (PAMP). Our study indicates that, indeed, β-1,6-glucan acts as a PAMP with immunostimulatory potential. As pointed by this reviewer, and similar to β-1,3glucans, the exposure of β-1,6-glucan is probably a key point in immune response. However, this investigation beyond the scope of this study, underway and will be presented in our future work.

- The Discussion would also benefit from an analysis of how β-1,6-glucan in Aspergillus fumigatus, which was largely elucidated by the same primary authors.

To our knowledge, β-1,6-glucan has never been identified, either by chemical analysis (PMID:10869365; PMID:36836270) or solid-state NMR (PMID:34732740), in the cell wall of *A. fumigatus*, although a homolog of *KRE6* is present in *A. fumigatus* but with unknown function.